# Assessing COVID-19 pandemic excess deaths in Brazil: Years 2020 and 2021

**Saditt Rocio Robles Colonia[1], Lara Morena Cardeal[1], Rogério Antonio de Oliveira[2], Luzia Aparecida Trinca[2] ***

**1** Research Program in Biometry, Unesp, Botucatu, São Paulo, Brazil, **2** Department of Biodiversity and Biostatistics, Institute of Biosciences, Unesp, Botucatu, São Paulo, Brazil

* luzia.trinca@unesp.br

**Data Availability Statement:** Original data are available from Ministério da Saúde, Governo do Brasil: https://opendatasus.saude.gov.br/dataset/sim-1979-2019 https://opendatasus.saude.gov.br/dataset/sim-2020-2021 We also provide the above

## Abstract

We estimated the impact of the COVID-19 pandemic on mortality in Brazil for 2020 and 2021 years. We used mortality data (2015–2021) from the Brazilian Health Ministry for forecasting baseline deaths under non-pandemic conditions and to estimate all-cause excess deaths at the country level and stratified by sex, age, ethnicity and region of residence, from March 2020 to December 2021. We also considered the estimation of excess deaths due to specific causes. The estimated all-cause excess deaths were 187 842 (95% PI: 164 122; 211 562, P-Score = 16.1%) for weeks 10-53, 2020, and 441 048 (95% PI: 411 740; 470 356, P-Score = 31.9%) for weeks 1-52, 2021. P-Score values ranged from 1.4% (RS, South) to 38.1% (AM, North) in 2020 and from 21.2% (AL and BA, Northeast) to 66.1% (RO, North) in 2021. Differences among men (18.4%) and women (13.4%) appeared in 2020 only, and the P-Score values were about 30% for both sexes in 2021. Except for youngsters (< 20 years old), all adult age groups were badly hit, especially those from 40 to 79 years old. In 2020, the Indigenous, Black and East Asian descendants had the highest P-Score (26.2 to 28.6%). In 2021, Black (34.7%) and East Asian descendants (42.5%) suffered the greatest impact. The pandemic impact had enormous regional heterogeneity and substantial differences according to socio-demographic factors, mainly during the first wave, showing that some population strata benefited from the social distancing measures when they could adhere to them. In the second wave, the burden was very high for all but extremely high for some, highlighting that our society must tackle the health inequalities experienced by groups of different socio-demographic statuses.

## Introduction

By the end of April 2023, Brazil's coronavirus disease (COVID-19) death toll was 701 494 (nearly 10% of the world confirmed deaths), putting the country among the most affected, behind the USA only [1]. After confirmation of the first cases, around the end of February 2020, the virus spread rapidly from the largest cities to the most vulnerable communities, reaching the whole country by March 2021 [2]. Several factors contributed to the far-beyond disaster in which the Brazilian federal government is blamed to be complicit [3–5]. The then

data aggregated at https://github.com/luziatrinca/COVID-19_excess_deaths_Brazil_2020_2021.

**Funding:** The first and second authors acknowledge research scholarships from Coordenação de Aperfeiçoamento de Pessoal de Nível Superior – Brasil (CAPES) – Finance Code 001.

**Competing interests:** The authors have declared that no competing interests exist.

President of the Republic chose to follow consistent scientific denialism and assumed a dismissive attitude towards the pandemic. In the middle of chaotic governance, by mid-April 2020, the Supreme Federal Court ruled that Federal, states and municipalities governments have concurrent competence to deliver on health issues and sub-national governments may implement non-pharmacological interventions against COVID-19 without the endorsement of the Federal Government [6, 7], even if more severe than those sanctioned by the Federal Government. Nonetheless, efforts were not saved from the federal sphere to undermine the public health responses to COVID-19, assuming a constant confrontation attitude, even for vaccine negotiation and acquisition, which delayed, provoked doubts and disrupted the vaccination process. The consequence was the most drastic. The misinformation spread, the lack of unity and leadership, added to the overcrowded cities, and the difficulties for low-income people accessing social security and engaging in physical distancing all have contributed to the worse [7–12]. Daily COVID-19 mortality reached figures around 1 000 from May to early September 2020. The second pandemic wave started in November, in which daily mortality soared, reaching even higher figures until July 2021, with the peak, at the end of March, showing more than 3 500 daily COVID-19 deaths [1, 13]. Allied with the already mentioned factors contributing to the virus spread, including mutation and new variants, there is huge regional diversity concerning health services access and cultural/socio-demographic factors across the country. Although Brazil allegedly has the largest public health system in the world, funded by government budgets, geographical and social inequalities strongly affect the capacity of the system and access to services [2, 14, 15].

Besides aggregating deaths by the new disease, the pandemic also affected the mortality pattern of other diseases due to changes in social conditions, individual behaviors and, importantly, lack of assistance due to a rather stressed health system. Deaths due to other infections, i.e. influenza, and external causes such as traffic/outdoor accidents and other injuries, are expected to be reduced, mainly in the first months of the pandemic, as a result of restricted social interactions and mobility. On the other hand, deaths due to other causes might have increased due to interruption of treatments, lack of preventive care and other factors. Another problem is the misdiagnose of the cause of death, which leads to under or over-reporting, especially for COVID-19, due to insufficient test capacity, and burdened health and recording systems [16, 17].

One approach for capturing the total impact, direct and indirect, is by estimating the excess deaths attributable to the pandemic, the difference between the observed and the predicted expectation or baseline deaths, over the same period, had the pandemic not happened [1, 18–20].

Excess deaths in Brazil have been broadly explored using different data sources, locations, periods, stratification factors and methods [9, 12, 14–16, 20–27]. The long-lasting pandemic and mortality data delay mean the theme requires constant updating. In this paper, we estimate all-cause excess death for 2020 and 2021 at several aggregation levels, such as in the whole country, by states and considering socio-demographic factors such as sex, age and race/color. For insights to understand excess and/or deficit, we also predicted excess deaths due to other specific group causes.

## Materials and methods

### Data

We obtained the publicly available data (2010–2021) from the Mortality Information System (SIM) [13], Brazilian Health Ministry on September 27, 2022 (for the datasets link see [13]). Each SIM record refers to an unidentified death, including the death date, some socio-

demographic factors and the primary cause. The data up to December 2020 are consolidated, while the data for 2021 are preliminary and are expected to change. We used data from week one, 2015, to week nine, 2020, for modeling the baseline deaths for the pandemic period and data from 2010–2019 for the evaluation of the prediction capabilities of the baseline models. Other Brazilian mortality data exist. Nonetheless, SIM is considered to have the best coverage. Based on 2016–2019 SIM data, IBGE (Instituto Brasileiro de Geografia e Estatística) estimated under-reporting rates ranging from 0.57% to 5.87% with major North and Northeast regions showing poorer performances [28]. Promotion of awareness and training of the registry and data processing personnel are improving the data quality and coverage of SIM over the years.

We grouped deaths by epidemiological week (US CDC definition) at several levels of stratification: country, federation unit, sex, age group, race/color and primary death cause. We coded age into five groups, namely 0–19, 20–39, 40–59, 60–79 and 80 or more years old. The factor race/color has five categories, namely White, Black, Brown, East Asian and Indigenous. We used this factor as a surrogate for hardship status (economic, education, opportunities access, including healthcare access) since, for historical reasons, the association between these two factors is very well known [29].

Brazil's territory is organized into five major regions, each sub-divided into several federation units: 26 states and the Federal District, hereafter states for simplicity. For death cause classification, we used the criteria of Santos *et al.* [20] and formed eight classes: COVID-19, Other Infectious Diseases, Neoplasms, Cardiovascular Diseases, Respiratory Diseases, Ill-Defined Causes, External Causes and Other Diseases. We note that all COVID-19 deaths recorded in the SIM used the ICD coding B-34.2 (no death has been recorded as U07.1 or U07.2 yet). The aggregated data set and program code can be retrieved from our online repository [30]. For data preparation and handling, we acknowledge enormous benefits from a publicly available program code [20].

## Methods

We used the linear mixed model (LMM) to predict baseline deaths. In the following, we present our reasoning in favor of this method.

The available methods broaden from simple five-year averaging to quite demanding models requiring census data [18, 19, 31–33]. Verbeeck *et al.* [19] brought to mind, apart from other problems, the data violations on assumptions of some of the approaches. A typical data violation is the implicit correlations of the historical death counts not incorporated in several models. Another point is that some methods fail to account for the specific-year mortality trend, i.e. averaging over the years without adjustments. The mixed models, devised for modeling grouped/correlated data [34], capture correlations within and heterogeneity among groups. The linear case was applied successfully to mortality data for two European countries [19]. The normality assumption is decisive for the simplicity and flexibility of the modeling, allowing the incorporation of a broader class of correlation structures, achieving predictions for specific groups, and attaining explicit expressions for the standard errors of the predictions. The last point is crucial to account for all uncertainty involved in the forecasts. Furthermore, it does not require information from census population projections. That is not necessarily the case when the modeling includes some non-linear link function and assumes some other non-normal distribution, as in the Generalized Linear Model (GLM) framework. Such approaches require more sophisticated methods, usually involving simulations, to provide accurate and reliable predictions [31, 35]. Yet, they do not include correlations.

The criticism of LMM is the underlying normality assumption, but since weekly mortality assumes large values, a good approximation is, generally, attained [9, 19].

Another possible approach is the application of the Generalized Estimation Equations (GEE) formulation [33], which does not require any distributional assumption and allows correlations among the observations. Its drawback is that it is not devised for predictions. The model estimates the mean profile, using grouped/correlated data, for the population from which the groups represent a sample [36, 37]. We may use the fitted equation to extrapolate it to future time points and obtain the point predictions. However, the precision of such predictions, considering only the uncertainty on the mean estimates, is not fair because it does not consider the uncertainty related to future observations. Palacio-Mejía *et al*. [33] recognized that the method did not consider the future observations' uncertainty. Failing accurate estimation of the standard errors of the baseline predictions could result in too narrow prediction intervals and lead to incorrect interpretations of excess deaths [35]. Nonetheless, we used this approach as a base for comparisons to the LMM approach, in terms of point predictions, applied to the Brazilian data.

**Modeling by the linear mixed model.** Using historical mortality data, for the LMM framework, each year is considered a cluster or group and weeks within a year are the observational units [19]. Terms of the Fourier Series (FS) capture the cyclic pattern and year random effects capture year trend, such that a basic model is

$$Y_{ix} = \beta_0 + \beta_1 \sin\left(\frac{2\pi x}{\mathcal{T}}\right) + \beta_2 \cos\left(\frac{2\pi x}{\mathcal{T}}\right) + b_i + \varepsilon_{ix} \tag{1}$$

where $Y_{ix}$ is the death count in year $i$ ($i = 1, 2, \cdots, 6$) and week $x$ ($x = 1, 2, \cdots, 52$, except for $i = 6$, the year 2020, for which $x = 1, 2, \cdots, 9$), $\beta$'s are fixed-effect parameters, $\mathcal{T}$ is the period (of the FS), $b_i \sim N(0; \sigma_b^2)$ is the year-specific random effect and $\varepsilon_{ix} \sim N(0; \sigma^2)$ is the random measurement error. As standard, $b_i$ and $\varepsilon_{ix}$ are assumed to be independent. It is possible to include other random effects to account for the heterogeneity of the regression parameters over the years, as can other relevant covariates of fixed effects.

The first documented COVID-19 death in Brazil occurred on March 12, 2020, allowing a non-pandemic period of nine weeks to predict the 2020-specific death trend, relaxing the requirement of census data to capture mortality rates. However, for 2021, the prediction of such a trend is impossible using Eq (1) and we would have to rely on the estimated population mean curve. To obtain more realistic forecasts for 2021 beyond the estimated population mean, which would underestimate the baseline deaths, we have included a linear term of time in the model. Thus, the more general LMM is

$$Y_{ix} = \beta_0 + b_i + \sum_{k=1}^{K}\left[(\beta_{1k} + b_{1ki})\sin\left(\frac{2k\pi x}{\mathcal{T}}\right) + (\beta_{2k} + b_{2ki})\cos\left(\frac{2k\pi x}{\mathcal{T}}\right)\right] + \beta_3 t + \varepsilon_{ix} \tag{2}$$

where $K$ is the number of terms needed in the FS, $b_{1ki}$ and $b_{2ki}$ are further random effects to account for year heterogeneity and $t = 1, 2, \cdots, 269$ indicates time points in the series of the data, starting in week 1, 2015 and ending in week nine, 2020. Using this type of model we predicted weekly mortality for 2020 and 2021, aggregated for all-cause deaths, at the country level and stratified by: state, sex, age group and race/color categories. We further obtained baseline values for primary death-cause groups. As the mixed model involves modeling mean and variance-covariance structures, changes in one part might impact the estimates of the other, and care should be taken for fitting a parsimonious model. We outline below the general steps we followed to select a parsimonious model for each stratification.

1. Fit a linear fixed-effects model using Ordinary Least Squares (OLS), considering the years as a blocking factor and the weeks as a qualitative factor. This model, with 58 parameters, is the completest mean model the data allow fitting. Denote it Model 0.

2. Search for some function of $x$ (week) that adequately captures the yearly cyclic pattern. In this step, the number of terms ($K$) required in the FS in Eq (2) should be fixed. To estimate the period $\mathcal{T}$ (or equivalently the frequency $\omega = \frac{2\pi}{\mathcal{T}}$), the approach indicated in [34] is used, that is, fit a non-linear model (non-linear least squares) to estimate the $\omega$ parameter. The non-linear model includes years as a blocking factor, a linear effect of time $t$ and $K$ fixed, say $K = 2$. Note that such a model captures the year effect but remains partially linear, requiring an initial value for $\omega$ only. With $\hat{\omega}$ given, fit the linear regression model that incorporates the functional form of $x$ and $t$ found previously. Note that such a model represents a considerate simplification of Model 0. Test for lack-of-fit of the simpler model. Under the evidence of lack-of-fit, increase $K$ and repeat the checking. In most cases, $K = 1$ or 2 resulted in parsimonious fits, but there were cases requiring $K = 3$. The final model in this step is Model 1.

3. Fit the LMM with the mean part found in step 2 and the random effects, one for accounting for years variability and one associated to each element of the FS in Eq (2). Note that once more than one random effect is included in the model, then we have a random vector $\mathbf{b}_i$, with some covariance matrix $\mathbf{D}$, such that $\mathbf{b}_i \sim N(\mathbf{0}; \mathbf{D})$ and the structure of $\mathbf{D}$ must be specified. We started with the most complex structure, i.e., the so-called unstructured pattern, which means any symmetric positive-definite matrix. The fitted model in this step is Model 2.

4. Simplify, if possible, the structure of $\mathbf{D}$ by declaring $\mathbf{D}$ diagonal, which means the random effects are uncorrelated. Denote Model 3 the most parsimonious model in this step.

5. Explore other models, possibly dropping random effects based on the magnitude of their variance component estimates and their standard errors, or performing some formal tests. The simpler model, not showing lack-of-fit compared to the more complex one, is kept. Call Model 4 the final model in this step.

6. Update Model 4 by incorporating serial correlation between observations within the same year. That is, the no-serial correlation model assumes $\boldsymbol{\varepsilon}_i \sim N(\mathbf{0}, \sigma^2 \mathbf{I})$. For serial correlation, $Var(\boldsymbol{\varepsilon}_i) = \mathbf{R}$, where $\mathbf{R}$ is a non-diagonal symmetric positive-definite matrix. Some usual possibilities are first-order auto-regressive (AR(1)), Gaussian and Spherical correlation patterns [34]. The most parsimonious model is Model 5.

7. Check for further simplification of the random part as some random effects in $\mathbf{b}_i$ might not be relevant after the serial correlation account. Keep the most parsimonious model (Model 6).

8. Check for simplification of the fixed effects (mean model) by applying a backward type selection.

9. Perform a detailed diagnostic analysis of the fit. In case of evidence of violations, some fix may be possible by following the recommendations in [38].

Applications of these steps allowed the selection of a model for baseline death predictions for each stratification we explored. For most cases, one term in the FS was enough to explain the seasonal mortality variation, but there were cases requiring two or three. In particular, for

External Causes of death, the FS function did not fit the data. We used a third-order polynomial instead. Often, the serial correlation structure was well-modeled by AR(1).

The estimated equation applied to weeks $x = 10, 11, \cdots, 53$, time points $t = 270, 271, \cdots 313$ and the predicted 2020-specific random effects provided the baseline death forecasts for the pandemic period of the year 2020. Similarly, for the year 2021 forecasts, weeks $x = 1, 2, \cdots 52$ and time points $t = 314, 315, \cdots, 365$ were applied, the difference being that the best predictions of 2021-specific random effects are null. For calculations of standard errors and prediction intervals, we used standard LMM theory (for details, see S1 File).

**Modeling by GEE.**   For the GEE formulation, the primary death cause stratum (seven in our case) defines the grouping. The model has the three components described below.

1. The link function (the natural choice is the log link since the response is death counts) related to the linear predictor

$$\log[E(Y_{ct})] = \log(\lambda_{ct}) = \mu_{ct} + \sum_{k=1}^{K}\left[\beta_{1ck}\sin\left(\frac{2k\pi t}{52}\right) + \beta_{2ck}\cos\left(\frac{2k\pi t}{52}\right)\right] + \beta_{3c}t \qquad (3)$$

where $\lambda_{ct}$ and $\mu_{ct}$ are, respectively, the expected death count and the intercept, in stratum $c$ ($c = 1, 2, \cdots, 7$) and time point $t$. The other terms follow definitions already stated around Eq (2), however, specific for stratum $c$.

2. The variance function: $Var(Y_{ct}) = \phi\lambda_{ct}$ whose form declares the variance follows the behavior of the over-dispersed Poisson model.

3. The correlation function specifies the correlation pattern among death counts at distinct time points. Here, we assumed the AR(1) structure.

We note that this is not the usual specification of the GEE approach used to model population mean profiles using a sample of groups or clusters because the linear predictor includes the specific effects for the grouping factor, and the fitted model estimates the mean profile group-specific. In this context, it should be so because there is no meaning in averaging the deaths (or log deaths) across death causes. By substituting $t = 270, 271, \cdots, 365$ in the fitted linear predictor of Eq (3) and applying exponentiation, a baseline mortality forecast was obtained for each time point and grouping. Aggregation across groups, following the methods in [33], resulted in baseline forecasts for all-cause deaths.

**Prediction accuracy.**   To assess the prediction accuracy of both modeling approaches, LMM and GEE, we used the usual measures related to prediction error (for details see S1 File). For this task, we used data from 2010–2019 such that for each year $i$ in turn, from 2015 to 2019 ($i = 1, 2, \cdots, 5$), we fitted each model being compared using data from the previous five-year history and the nine first weeks of year $i$. Then, for a year $i$, we obtained forecasts for weeks 10, 11, $\cdots$, 52 and calculated the prediction error as the difference between the actually observed and the predicted death counts. We conducted this investigation only for the all-cause deaths at the country level.

**Excess deaths.**   The excess death estimate for each time point is the observed death count minus the predicted baseline. For year summaries and fairer comparisons between strata, we used the P-Score (per capita of excess death in percentage) [39] and the $\text{Ratio}_{EC}$ (ratio excess by COVID-19 deaths) [20]. The year-accumulated P-Score is defined as $P - \text{Score}_i = 100 \times \frac{E_i}{\tilde{y}_i}$ where $E_i$ is the accumulated excess and $\tilde{y}_i$ is the accumulated baseline forecast deaths for year $i$. The year-accumulated ratio excess is defined as $\text{Ratio}_{ECi} = \frac{E_i}{y_i^c}$ where $y_i^c$ is the annual COVID-19 confirmed deaths. The $\text{Ratio}_{ECi}$ measures the COVID-19 pandemic effect on deaths attributable to other causes. Values below one mean that COVID-19 surpassed excess deaths such that

there was a deficit in reporting deaths due to other causes, while values above one mean extra deaths due to other causes beyond COVID-19 amounted.

**Computational resources.** We used R [40] for all computations (packages: `nlme` [41] for the LMM, `varTestnlme` [42] for adjusting p-values when necessary, `geepack` [43] for GEE model, `ggplot2` [44] for the plots). For the fitting diagnosis, we used the R function `lmmdiagnostic` [38].

## Results

### LMM versus GEE

Firstly we show the results comparing the performances of the best fittings obtained following the two modeling alternatives presented in the methodology. We concentrate on the forecasting of all-cause baseline deaths at the country level. Fig 1 shows the death count series from the first week of 2015 to week nine of 2020 and the fitted curves. Both models underestimate the yearly peak, with LMM showing a somewhat better performance. LMM also captures the lower spikes that appear consistently at the end/beginning of each year. Table 1 presents summaries of prediction accuracy measures when using the fitted models to predict the number of deaths for weeks from 10 to 52 for years from 2015 to 2019. LMM and GEE present very similar performances that are, unsurprisingly, much superior to the five-year average method.

The reasonable agreement between the LMM and GEE in point predictions can also be seen in Table 2, although there is a tendency for lower baseline forecasts from the GEE. Given the flexibility of the LMM, allowing explicit formulae for uncertainty estimation around point predictions, we will concentrate, in the following sections, on LMM results.

### Excess deaths at the country level

Detailed results from our step-by-step strategy to model the baseline deaths are presented in S2 File. The fitted baseline model for the year 2020 ($i = 6$), Eq (4), is

$$
\tilde{y}_{6x} = [24539.37 + 8.95] + 6.79 \times t + [-1170.28 + 20.75] \times \sin(0.16 \times x) + \\
[-859.39 + 0.73] \times \cos(0.16 \times x)
\tag{4}
$$

where $x = 1, 2, \cdots, 53$ and $t = 260, 261, \cdots, 313$. The second figure within each set of square brackets is the prediction of the 2020-specific effect. Diagnostics graphs did not reveal any concern about violations of the assumptions of the model (see S1 Fig). For coefficient estimates, their standard errors and estimates of all other model parameters see S1 Table. For 2021, the equation is the same except that year-specific effects are set to zero and $x$ runs from 1 to 52 and $t$ from 314 to 365. The great impact if we predict for 2020 or 2021 is, therefore, concerning the uncertainty around the predictions (see S1 File) as we will show in the graphs.

Fig 2 shows the observed weekly all-cause mortality for week 1 of 2015 to week 52 of 2021, the baseline forecasts for both years, and the 95% PI for expected mortality plus reported COVID-19 deaths (forecast + COVID-19).

Excess has occurred over the whole period since the surge of the first wave in 2020 and reached alarming figures in the second wave in 2021. In the beginning, excess indicates that more deaths occurred beyond COVID-19. From week 13 to around week 22, 2020 (last week of June), excesses agreed well with COVID-19 as the primary cause. From week 22 to approximately week 32, 2020 (beginning of August), COVID-19 exceeded deaths from all causes, i.e., deaths due to other causes were smaller than expected. A peak of excess deaths occurred again in week 41 (around 10 October). From week 44 until the end of the year, deaths increased with the second wave, and excesses agree well with reported COVID-19 deaths. In 2021, the spread

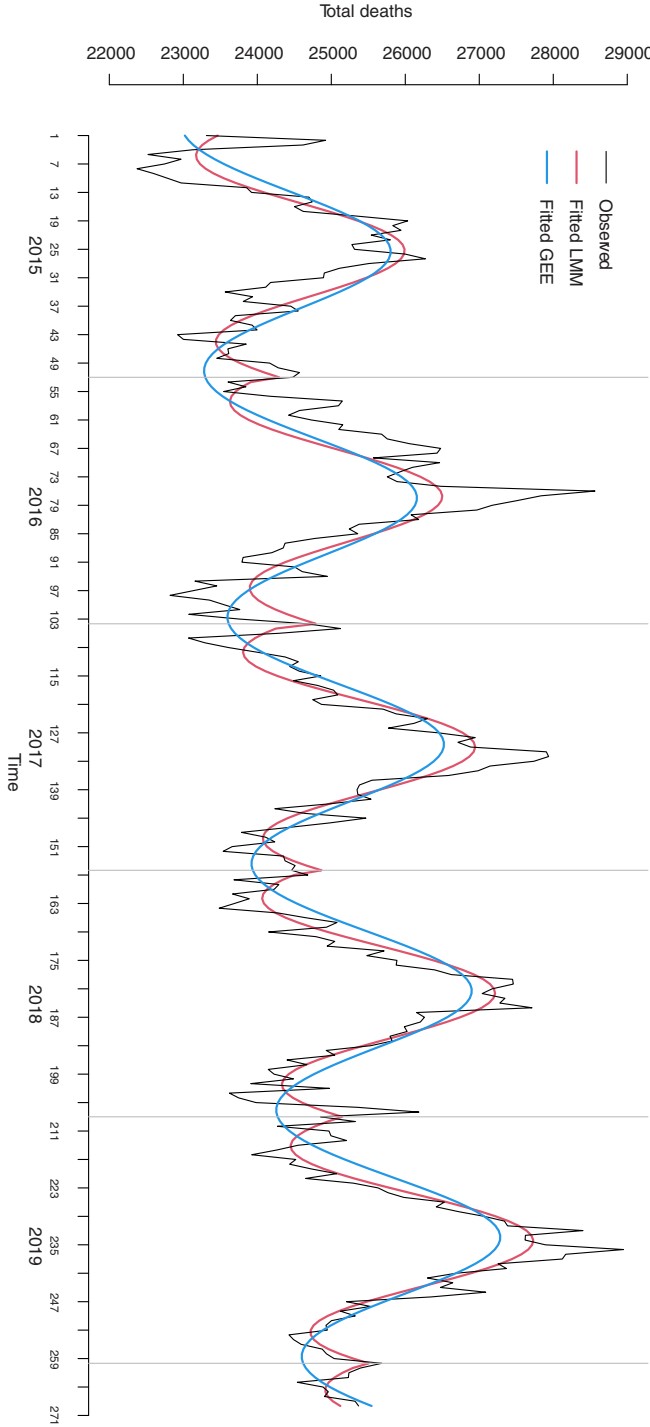

**Fig 1. Weekly all-cause mortality in Brazil from week 1, 2015 to week nine, 2020.** Recorded and fitted mortality by the LMM and the GEE approach.

**Table 1. Summaries for accuracy measures under LMM and GEE for forecasting death numbers in future weeks of the year, using data from the previous five-year period up to the ninth week of the current year.**

| Model | Summary | Accuracy measures[a] | | | | | |
|---|---|---|---|---|---|---|---|
| | | ME | MAE | RMSE | RRMSE | MAPE | MPE |
| LMM | *min* | -371.52 | 389.69 | 544.88 | 0.32 | 1.45 | -1.51 |
| | *mean* | 59.92 | 614.14 | 748.92 | 0.45 | 2.42 | 0.15 |
| | *median* | 189.21 | 521.45 | 622.90 | 0.37 | 2.07 | 0.65 |
| | *max* | 384.49 | 961.88 | 1145.50 | 0.69 | 3.77 | 1.50 |
| GEE | *min* | -447.29 | 515.05 | 653.42 | 0.38 | 1.95 | -1.83 |
| | *mean* | -27.31 | 664.32 | 794.63 | 0.48 | 2.62 | -0.21 |
| | *median* | 59.98 | 646.64 | 763.45 | 0.46 | 2.62 | 0.13 |
| | *max* | 288.53 | 919.68 | 1091.20 | 0.66 | 3.61 | 1.00 |
| Five-year Average | *min* | 1120.64 | 1131.94 | 1244.81 | 0.74 | 4.39 | 4.35 |
| | *mean* | 1496.20 | 1498.46 | 1640.46 | 0.99 | 5.86 | 5.85 |
| | *median* | 1525.41 | 1525.41 | 1593.01 | 0.96 | 5.95 | 5.95 |
| | *max* | 1865.60 | 1865.60 | 2136.65 | 1.29 | 7.25 | 7.25 |

a: ME: mean error; MAE: mean absolute error; RMSE: root mean squared error; RRMSE: relative root mean squared error; MAPE: mean absolute percentage error and MPE: mean percentage error.

of variant Gamma before vaccination began, associated with further relaxation of social distancing, impacted mortality significantly. Only by mid of July 2021, death numbers lowered down to the level of the worst period of the previous year. For the occasion of the peak, around weeks 12–13 (end of March), only 2% of the country's population was fully vaccinated.

**Table 2. Reported, expected and estimated excess/deficit deaths by primary selected cause, accumulated for two periods, weeks 10–53, 2020 and weeks 1–52, 2021.**

| Year | Cause | Reported | Expected | | Excess | | 95% PI[a] | | P-Score[a] |
|---|---|---|---|---|---|---|---|---|---|
| | | | LMM | GEE | LMM | GEE | | | |
| 2020 | All-cause | 1 351 320 | 1 163 478 | 1 136 629 | 187 842 | 189 096 | 164 122 | 211 562 | 16.1 |
| 2020 | COVID-19 | 214 620 | | | | | | | |
| 2020 | Cardiovascular | 302 999 | 313 349 | 306 593 | -10 350 | -3 594 | -17 757 | -2 943 | -3.3 |
| | Other Diseases | 269 045 | 271 267 | 265 817 | -2 222 | 3 228 | -9 926 | 5 483 | -0.8 |
| | Neoplasms | 191 404 | 204 005 | 199 772 | -12 601 | -8 368 | -14 654 | -10 548 | -6.2 |
| | Respiratory | 125 941 | 140 921 | 137 907 | -14 980 | -11 966 | -21 348 | -8 613 | -10.6 |
| | External | 122 569 | 124 901 | 119 499 | -2 332 | 3 070 | -6 522 | 1 859 | -1.9 |
| | Ill-defined | 79 235 | 64 145 | 61 375 | 15 090 | 17 860 | 10 927 | 19 254 | 23.5 |
| | Other Infect. | 45 507 | 47 304 | 46 365 | -1 797 | -858 | -2 817 | -777 | -3.8 |
| 2021 | All-cause | 1 821 737 | 1 380 689 | 1 382 939 | 441 048 | 438 798 | 411 740 | 470 356 | 31.9 |
| 2021 | COVID-19 | 420 193 | | | | | | | |
| 2021 | Cardiovascular | 377 828 | 368 705 | 368 135 | 9 123 | 9 693 | -874 | 19 119 | 2.5 |
| | Other Diseases | 337 057 | 326 048 | 328 318 | 11 009 | 8 739 | 2 435 | 19 582 | 3.4 |
| | Neoplasms | 233 344 | 246 810 | 248 149 | -13 466 | -14 805 | -15 743 | -11 188 | -5.5 |
| | Respiratory | 142 975 | 165 266 | 164 740 | -22 291 | -21 765 | -29 439 | -15 144 | -13.5 |
| | External | 147 003 | 150 870 | 143 440 | -3 867 | 3 563 | -15 713 | 7 979 | -2.6 |
| | Ill-defined | 100 645 | 80 508 | 73 985 | 20 137 | 26 660 | 11 091 | 29 183 | 25.0 |
| | Other Infect. | 62 692 | 55 605 | 56 172 | 7 087 | 6 520 | 4 794 | 9 379 | 12.7 |

*a*: based on the LMM model.

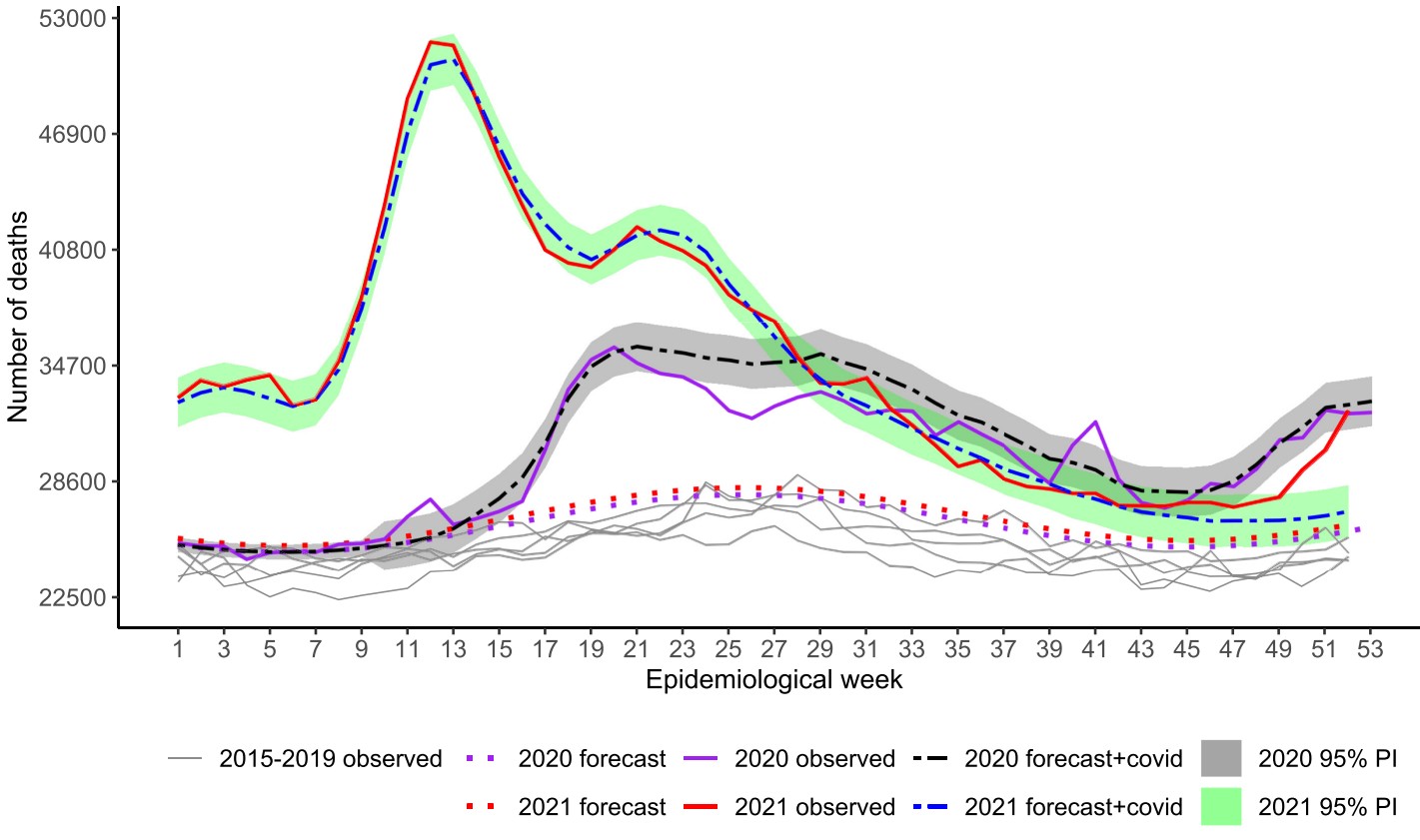

**Fig 2. Weekly all-cause mortality in Brazil from week 1, 2015 to week 52, 2021.** Recorded and baseline mortality forecast by the LMM and the forecast plus observed COVID-19 deaths including 95% prediction intervals (PI) for 2020 and 2021.

Vaccination for those classified as high-risk groups, e.g., elderly, and health workers, started very slowly on January 18, 2021 [45, 46]. As vaccination advanced, mortality tended to approach the expected figures, though, by December, excesses increased again and reached the previous year's levels.

Accumulated deaths in each year showed 214 620 COVID-19 and 187 842 estimated excess deaths (95% PI: 164 122 to 211 562; $Ratio_{EC}$ = 0.88) for weeks 10–53, 2020, and 420 193 COVID-19 and 441 048 estimated excess deaths (95% PI: 411 740 to 470 356; $Ratio_{EC}$ = 1.05) for weeks 1–52, 2021 (Table 2). The P-Score estimates pointed out 16.1% and 31.9% more deaths than expected in the pandemic 2020 and 2021, respectively.

Table 2 also includes estimates accumulated by specific death causes (model parameter estimates in S1 Table), which, along with Fig 3 show that excesses and deficits occurred for most causes, at different points in time, except for Ill-defined Causes showing substantial deaths since the pandemic started, totaling 15 090 (23.5%) and 20 137 (25.0%) excesses in 2020 and 2021, respectively.

For Respiratory, excesses occurred essentially at the beginning of the pandemic up to around week 20 (end of May 2020). After that, the impact was negative, with deficits until week 41 (early October) with the surge of a sudden spike. Overall, the effect was negative, showing a deficit of −14 980 (−10.6%). By week 12 (end of March) of 2021, we saw a peak, followed by decreasing figures lower than expected during almost the rest of the period, except

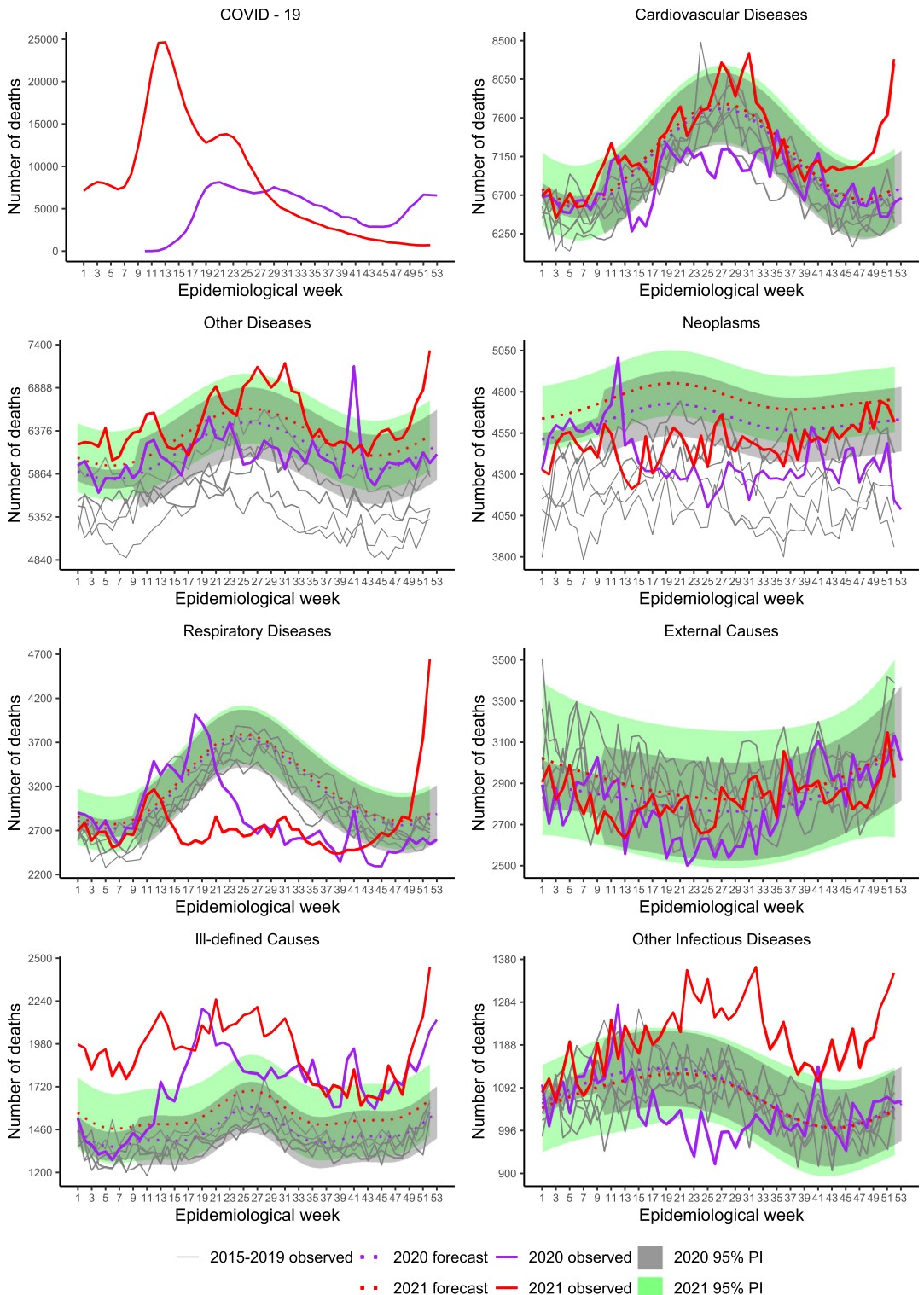

**Fig 3. Weekly mortality in Brazil, stratified by death cause, from week 1, 2015 to week 52, 2021.** Recorded and baseline mortality forecast by the LMM including 95% prediction intervals (PI) for 2020 and 2021.

that, by the end of November, there was a sharp increase. Overall, the balance was negative, showing a deficit of −22 291 (−13.5%). Mortality under Other Infectious Diseases showed a similar pattern to Respiratory in the year 2020, although on a smaller scale, with deficit deaths occurring during the weeks in the middle of the year (overall deficit of −1 797 and P-Score = −3.8%). However, in 2021, expressive mortality occurred, resembling the pattern of Ill-defined Causes (overall excess 7 087 and P-Score = 12.7%).

Deficit deaths due to Cardiovascular Diseases occurred from March to August 2020, the period of tighter control measures. Overall the balance was −10 350 (−3.3%). In 2021, the mortality pattern followed the expected, except for the significant increase by the end of the year. Further analysis, when 2021 consolidated data are available, should be performed to confirm such an unexpected increase.

A peak stands out for Neoplasms at the beginning of the pandemic, followed by deficits for the rest of 2020. The year balance is −12 601 (−6.2%). The deficits persisted for most of the year 2021 with an overall balance of −13 466 (−5.5%). Note, however, that the baseline curves are, perhaps, overestimating, mainly for 2021. Such a pattern is due to the death series that showed a strong positive slope over time.

Overall, External Causes did not suffer the expected impact with the reduction of traffic and outdoor movements during restrictions, perhaps because many people were unable to comply with the recommendations.

From these inspections, we note that the first peak of excess deaths (week 12) in Fig 2 is related mainly to Ill-defined Causes, Neoplasms, Respiratory and Other Infectious Diseases. The peak at week 41, 2020, is related to these causes, except for Neoplasms.

## Excess deaths by state

Excess deaths occurred in all states, but with enormous heterogeneity across the country. Fig 4 (left and center) and S2 Table present the accumulated statistics for each state by the period. States in the North and Central West regions have P-Score values (left panel of 4) well above the value for Brazil (vertical lines), in both years, with Amazonas, Mato Grosso and the Federal District, with values ranging from 29.1–38.1% being the highlights in 2020. On the other hand, the states in the South are the highlights for the low percentages of excess death in that year, markedly Rio Grande do Sul with 1.4%. For 2021, the figures increased for all states, with Rondônia and Amazonas suffering alarming mortality, above 50%. Surprisingly, all states in the Northeast region have lower percentages than the global level, which, however, is quite large (31.9%).

Several states suffered more excess deaths than COVID-19 in both years (central panel in Fig 4), mainly states in the Northeast, Central West and North regions. Deficit death was the rule, markedly in the South and Southeast regions, in the first year, with the lowest ratio (0.11) for Rio Grande do Sul. However, for the second wave, in 2021, most states showed ratios above 1, indicating that excess surpassed COVID-19 deaths. The minor figures (around 0.80) were for Roraima and Acre (in the North) and for Rio Grande do Sul (in the South). Detailed results along time, shown in S2–S6 Figs, indicate that, in general, the states that coped better with the pandemic in the first year (as highlighted above) showed deficit deaths during the first wave, suggesting that their population afforded better care for other diseases and, perhaps, afforded better compliance to isolation recommendations. However, cause-diagnostic mistakes and COVID-19 under-reporting, mainly in remote states, could also contribute to the figures.

In 2021, in most states, we cannot detect deficit deaths as the pandemic hit so badly the whole country, and it was much more difficult for the population to practice isolation, except

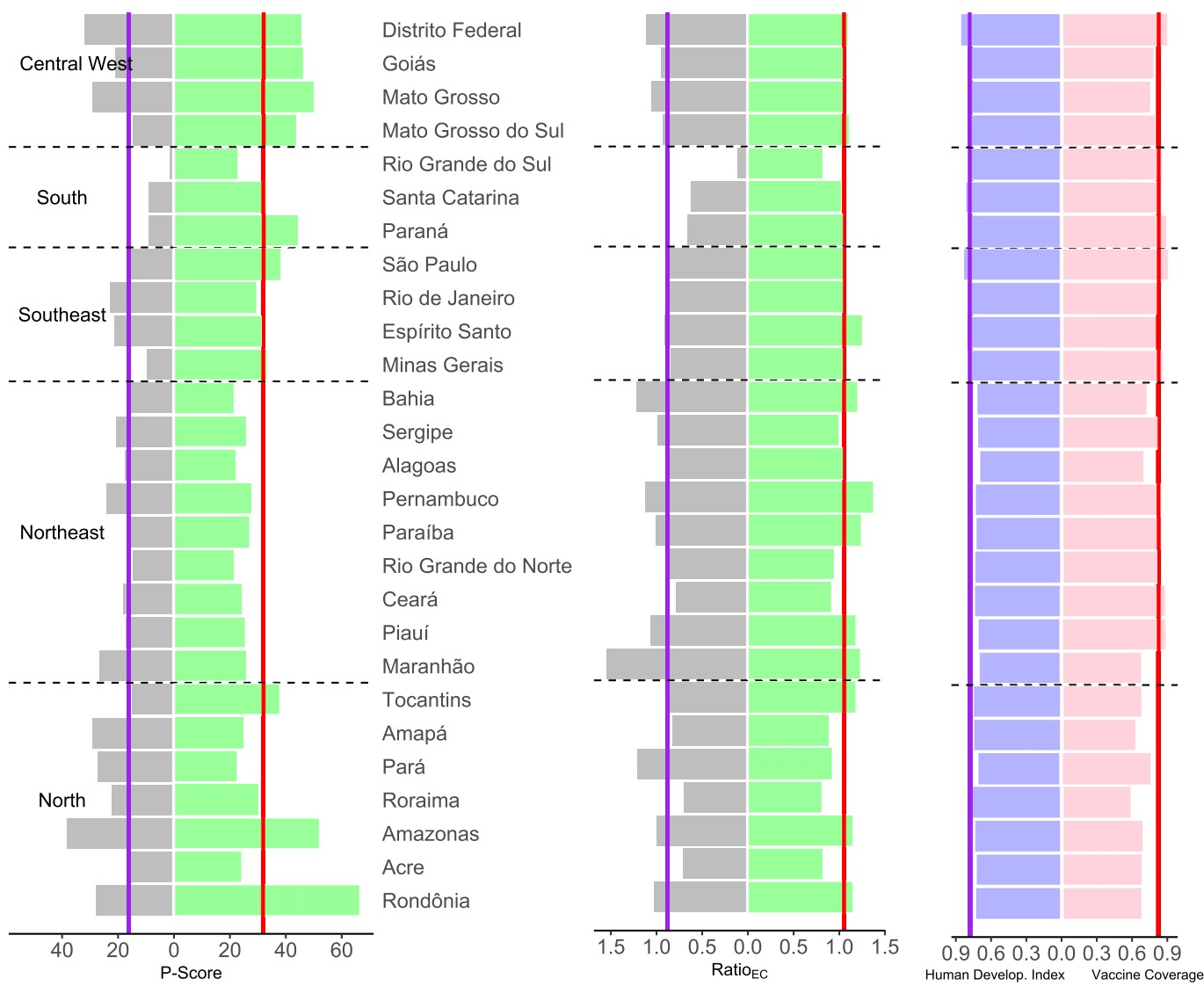

**Fig 4. P-Score (left panel) and Ratio$_{EC}$ (center panel) for each state, accumulated for two periods, weeks 10–53, 2020 (left in both panels) and weeks 1–52, 2021 (right in both panels).** Human Development Index (for year 2017) and Vaccination Coverage (2 doses), population share by 21 January 2023, for each state (right panel). The vertical lines mark the values at the country level.

for Rio Grande do Sul, which shows a consistent pattern of deficit deaths. Still, some states showed notable positive discrepancies between forecast+COVID-19 deaths and reported deaths again, mainly in the North, Northeast and Central-West regions. The drivers of the high heterogeneity across the country are many interacting factors: socio-economic and cultural characteristics, health system capacity (infrastructure of hospital care, laboratories), access to healthcare services, and local policy responses, to cite some. In an attempt to throw some light on the understanding of the regional disparities, in the right panel of Fig 4, we present the Human Development Index (HDI) [47] at left and the share of the vaccinated population, at least two doses, as for 20 February 2023 [48] at right. The North region is disadvantaged concerning the HDI scores and health system capacity, reflected here, we could argue, by the low vaccination coverage share. It is the largest Brazilian region (almost 50% of

the country's territory) where transport options are scarce, and access to healthcare services is poor. Manaus, the capital of Amazonas, experienced a collapse of its health system at the end of 2020/beginning of 2021, leading to very high mortality [49]. However, for the strongly impacted Central West region, but with high HDI, the above reasoning does not seem to apply, although we should keep in mind that HDI scores vary within each state, and this region is also vast in terms of area.

## Excess deaths according to sex, race/color and age

The SIM data is incomplete for sex, age and race/color, and so, before presenting the results on excess deaths according to these factors, we present their missing pattern. For sex and age, the percentages per year were small, just about 0.05% and 0.25% or less, respectively. Therefore, we do not expect biased results for these factors.

Fig 5(A) and Table 3 (for the parameter estimates of the models see S3 Table) show that for females, excess (13.4%) was smaller than for males (18.4%). Nonetheless COVID-19 surpassed excess during several weeks in the middle of 2020, producing $Ratio_{EC} = 0.76$ for females, while for males, differences were smaller and for shorter period, resulting $Ratio_{EC} = 0.96$. However, sex differences disappeared in 2021 with ratios of 1.03 and 1.07 and P-Score of 30.9% and 33.0% for females and males, respectively.

The missingness for race/color appeared in outstanding percentages, ranging from 2.07 to 4.50%, with the earlier years presenting the larger shares. These figures are substantial since they exceed the share of Indigenous and East Asian ancestries, which together, represent about 1.5% of the country's population. The missing distribution across death-cause, sex, and age showed a homogeneous pattern so that we could consider it at random for these factors. However, across states, we found very high missing percentages in Alagoas (11.9 – 18.1%) and Bahia (4.1 – 9.9%), in the Northeast, and Espírito Santo (11.6 – 14.2%) and Minas Gerais (1.6 – 8.8%) in the Southeast region. While Bahia and Minas Gerais showed a clear decrease in the figures over the years, that was not the case for Alagoas and Espírito Santo. The problem seems more related to awareness of careful data recording than to places' remoteness or availability of resources (or any other factor we could study). According to the last Brazilian census (2010), the shares of the joint categories Brown and Black to these state populations were 87% in Bahia, 67% in Alagoas, 57% in Espírito Santo and 53% in Minas Gerais [50]. For the country, the share of Black+Brown is about 50%. Consequently, we expect that these categories are under-represented in our analysis.

Fig 5(B) shows deficit deaths for the White category only during the first wave when recommendations for social distancing were in place. For the East Asian, Brown and Black categories, excess surpassed COVID-19 deaths at several periods, including 2021 for the Black and the East Asian groups. The $Ratio_{EC}$ values ranged from 0.70 (White) to 1.25 (East Asian) in 2020 and from 0.91 (Indigenous) to 1.27 (East Asian) in 2021. The P-Score varied from 12.4% (White) to above 22.0% (22.7% for Indigenous, 22.9% for Black and 28.6% for East Asian) and from 16.7% (Indigenous) to above 34.0% (34.3% for White, 34.7% for Black and 42.5% for East Asian), in 2020 and 2021, respectively, showing the Black and East Asian populations suffered severely from both viewpoints, higher per capita mortality and excess deaths due to other causes beyond COVID-19. We note the high variation in the death counts for Indigenous and East Asian groups over the years. We could suspect poor quality data recording for the first, but the reasons are unclear for the second since most East Asian descendants live in the Southeast region. We should keep in mind the data sparsity for these two groups, though. Given the unexpected pattern for this category, we investigated the historical deaths for each race/color category stratified by death cause (see S7–S11 Figs). For the East Asian group, we detected

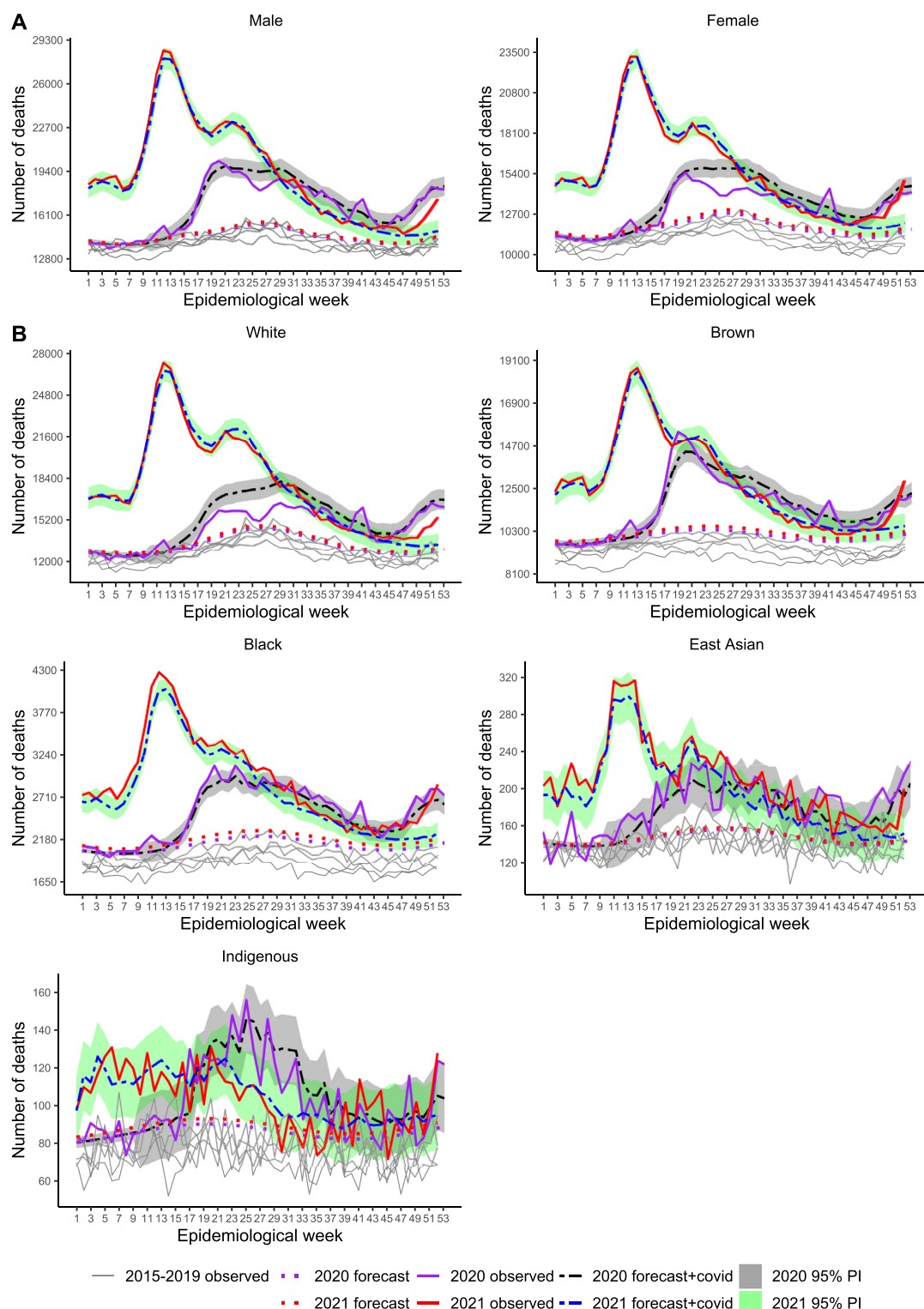

**Fig 5. Weekly mortality in Brazil, stratified by sex (A) and race/color (B), from week 1, 2015 to week 52, 2021.** Recorded and baseline mortality forecast by the LMM including 95% prediction intervals (PI) for 2020 and 2021.

**Table 3. Reported, expected and estimated excess/deficit total deaths and deaths by sex, race/color and age group, accumulated for two periods, weeks 10–53, 2020 and weeks 1–52, 2021.**

| Year | Strata | Reported | | Expected | Excess | 95% PI | | P-Score | Ratio$_{EC}$ |
|------|--------|----------|-----------|----------|--------|--------|--------|---------|--------------|
| | | COVID-19 | All-cause | | | | | | |
| 2020 | **Sex** | | | | | | | | |
| | Male | 122 689 | 759 890 | 641 705 | 118 185 | 104 918 | 131 452 | 18.4 | 0.96 |
| | Female | 91 917 | 590 894 | 521 016 | 69 878 | 59 672 | 80 083 | 13.4 | 0.76 |
| | **Race/Skin Color** | | | | | | | | |
| | White | 104 606 | 662 424 | 589 222 | 73 202 | 60 350 | 86 054 | 12.4 | 0.70 |
| | Brown | 82 194 | 524 902 | 444 532 | 80 370 | 68 296 | 92 444 | 18.1 | 0.98 |
| | Black | 18 735 | 115 357 | 93 846 | 21 511 | 19 941 | 23 081 | 22.9 | 1.15 |
| | East Asian | 1 474 | 8 344 | 6 488 | 1 856 | 1 608 | 2 104 | 28.6 | 1.26 |
| | Indigenous | 978 | 4 678 | 3 812 | 866 | 679 | 1052 | 22.7 | 0.89 |
| | **Age (years)** | | | | | | | | |
| | 00–19 | 1 029 | 49 137 | 52 573 | −3 436 | −5 357 | −1 515 | −6.5 | <0 |
| | 20–39 | 7 835 | 103 293 | 93 520 | 9 773 | 5 346 | 14 200 | 10.5 | 1.25 |
| | 40–59 | 40 474 | 252 476 | 207 446 | 45 030 | 39 417 | 50 643 | 21.7 | 1.11 |
| | 60–79 | 104 841 | 542 432 | 451 198 | 91 234 | 83 006 | 99 462 | 20.2 | 0.87 |
| | 80+ | 60 416 | 401 964 | 356 840 | 45 124 | 35 766 | 54 482 | 12.6 | 0.75 |
| 2021 | **Sex** | | | | | | | | |
| | Male | 233 370 | 1 008 941 | 758 390 | 250 551 | 231 089 | 270 013 | 33.0 | 1.07 |
| | Female | 186 773 | 812 091 | 620 512 | 191 579 | 178 991 | 204 166 | 30.9 | 1.03 |
| | **Race/Skin Color** | | | | | | | | |
| | White | 235 836 | 938 381 | 698 944 | 239 437 | 222 914 | 255 961 | 34.3 | 1.02 |
| | Brown | 138 706 | 671 826 | 530 178 | 141 648 | 123 177 | 160 120 | 26.7 | 1.02 |
| | Black | 32 028 | 152 582 | 113 235 | 39 347 | 37 565 | 41 130 | 34.7 | 1.23 |
| | East Asian | 2 576 | 10 945 | 7 680 | 3 265 | 2 947 | 3 582 | 42.5 | 1.27 |
| | Indigenous | 845 | 5 386 | 4 616 | 770 | 522 | 1 018 | 16.7 | 0.91 |
| | **Age (years)** | | | | | | | | |
| | 00–19 | 1 389 | 58 564 | 61 410 | −2 846 | −6 215 | 524 | −4.6 | <0 |
| | 20–39 | 26 628 | 140 942 | 106 549 | 34 393 | 27 349 | 41 438 | 32.3 | 1.29 |
| | 40–59 | 121 892 | 382 445 | 240 516 | 141 929 | 133 086 | 150 772 | 59.0 | 1.16 |
| | 60–79 | 190 824 | 736 079 | 540 258 | 195 821 | 185 381 | 206 261 | 36.2 | 1.03 |
| | 80+ | 79 425 | 501 307 | 427 705 | 73 602 | 61 506 | 85 698 | 17.2 | 0.93 |

peaks for Other Diseases, External Causes, Ill-defined Causes and Respiratory Diseases in 2020, while for 2021, we additionally detected peaks for Neoplasms. Notably, the plots do not reveal deficit deaths for any cause of death in this group. The last panel in Fig 5 shows an early death peak (week 5 of 2020) related to Ill-defined Causes.

Fig 6 shows the results stratified by age group. Deficit deaths prevailed in the group younger than 20 years old in 2020 (−6.5%) and 2021 (−4.5%). We note slight indication of deficit deaths, during the first wave, for the group 20–39 years old, but that disappeared from week 36, showing several peaks after September, 2020 and for 2021, so that the overall balance was positive, 10.5% in 2020 and 32.3% in 2021. This group presented the largest ratio, 1.25 in 2020 and 1.29 in 2021, indicating excess deaths due other causes than COVID-19. Note that, compared to the observed trajectories for the previous years, the baseline curves for 2020 and 2021 are lower. That is explained by a strong reduction tendency in the number of deaths, especially from 2016 to the weeks before the pandemic. Adults belonging to the 40–59 years old group show excess deaths surpassing COVID-19 practically over the two years, with ratio of 1.11 and

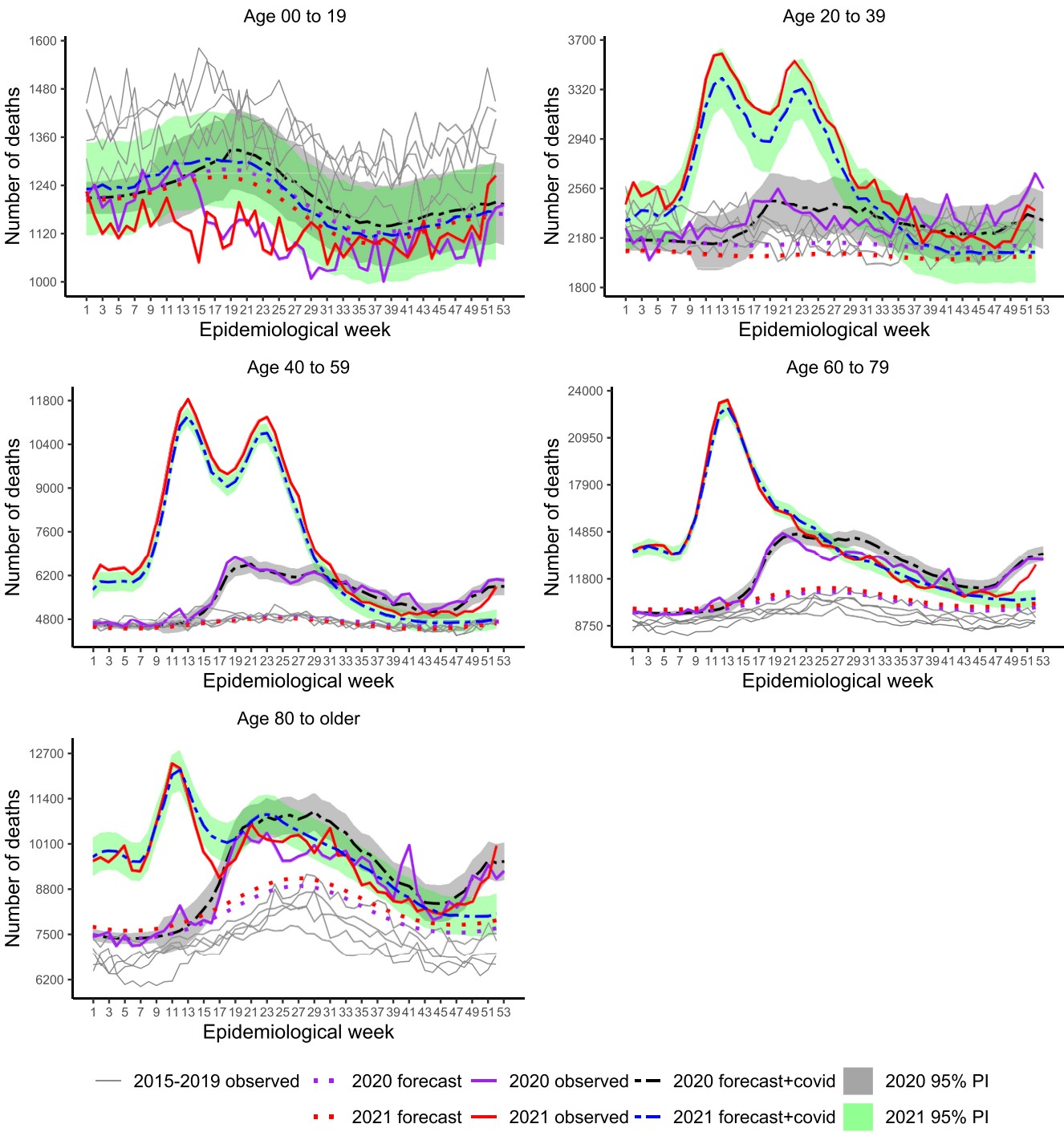

**Fig 6. Weekly mortality in Brazil, stratified by age group, from week 1, 2015 to week 52, 2021.** Recorded and baseline mortality forecast by the LMM including 95% prediction intervals (PI) for 2020 and 2021.

1.16 and P-Score of 21.7% and 50.9%, for 2020 and 2021, respectively. For the two oldest groups, the overall balance was positive as well, although there was a period of deficit deaths by the middle of the year 2020. The group of 80 years or older, experienced deficit deaths in 2021 as well, mainly after the start of vaccination, however that was canceled out by the steep increase by the end of 2021. The ratios for this group were smaller than 1 in both years (0.75 and 0.93) indicating that, overall excess deaths had COVID-19 recorded as the main cause.

## Discussion

In this paper, we accessed the impact of the COVID-19 pandemic on Brazilian mortality over the years 2020 and 2021, by estimating excess deaths stratified by several factors.

Since the COVID-19 pandemic declaration in March 2020, excess deaths in Brazil amounted to 16.1% and 31.9% more than expected in 2020 and 2021, respectively. For 2020, our results unfold similar patterns published elsewhere (see [20, 27], for example), although estimates vary because of the different modeling approaches, data updating, stratification factors and period considered. While we included only the pandemic weeks of 2020, other authors presented accumulated statistics for the entire year. Reported P-Score values are 13.7% (using GLM [20]), 14% (using an auto-regressive model [27]), 19% (using averaged previous five-year mortality rate [51]) and 25% (using averaged previous five-year death counts [26]). As foreseen, using averaged statistics results in underestimated expected deaths and thus, in overestimated excess. As also reported in other studies [16, 17, 20], our investigation highlights the enormous heterogeneity across Brazil, showing that the South was less impacted (1.4–9.0%), while the North and Central-West had the highest P-Score values approaching 30% excess in several of them and almost 40% in the Amazonas, in 2020.

Except for one study [32], whose work is almost continuously updated online and currently presents statistics up to 27 April 2023, we are not aware of other publications that accessed 2021 excess, for as long a period as we have. Our point estimates for the all-cause excess deaths and those from Karlinsky *et al.*'s approach [32] are in good agreement in both years (187 842 against 190 311 in 2020 and 441 048 against 439 854 in 2021, once we adjust for the same dataset). In addition, our work investigates further excess deaths stratified by socio-demographic factors and the causes of death.

Nucci et al. [27] estimated 40% excess deaths up to week 14 of 2021, comprising the most critical period in terms of daily COVID-19 deaths in Brazil, when only about 8% of the population had at most the second vaccine dose [1]. That is larger than our estimate for the whole year (31.9%) but expected with the advance of vaccination since, by the end the year, 77% of the population was at least partially vaccinated. Across the country, our P-Score estimates ranged from around 20% (in four states, two in the Northeast and one in the South) to above 50% in two states in the North. While several factors resulting in societal inequalities are likely to contribute to regional differences, for Brazil, we should add the failure of the country to deliver, as a whole, the awareness of disease severity, the relevance of non-pharmacological measures and the benefits of vaccination. Although caution is needed because death misdiagnosis is more likely in the poorest and remotest locations, the results reveal economic power does not explain much of the regional heterogeneity of excess deaths across the country.

Our results on deficit deaths from causes other than COVID-19 agree with those from other studies. Cause-specific expected mortality indicated deficits for Respiratory, Other Infectious Diseases, Neoplasms and Cardiovascular Diseases, in the year 2020, as also pointed out in other studies [20, 27, 51], although some of them [27, 51] used different death-cause grouping. With the social distancing recommendations, we expect lower exposure to risk factors

associated with the first two, while deficits for Neoplasms and Cardiovascular diseases might be explained by the evidence that people suffering from these illnesses were at high risk for serious COVID-19 conditions and died from COVID-19 in the first wave [23, 27, 52–54]. While these arguments are plausible, we should keep in mind that misdiagnosis is always an issue, mainly in deaths that happened outside hospitals and clinics and in regions without the capacity for proper diagnosis [22]. Of these diseases, only Respiratory and Neoplasms maintained the deficit pattern in part of 2021.

Excesses deaths beyond COVID-19 were typical at the beginning of the pandemic, about the end of the first wave, and at the end of 2021, a signal of possible death cause misreporting [17, 20, 55, 56]. Excess deaths due to Ill-defined Causes occurred in both years with Other Infectious Diseases added in 2021 which show, as well, a steep increase of deaths from Respiratory and Cardiovascular diseases at the end of the year. Once definitive data for 2021 are available, it is crucial to reassess them to confirm or disregard such patterns.

Studies around the world [20, 23, 57–59] reported COVID-19 mortality affects more males than females, although some authors did not find relevant differences once baseline was considered [60]. Our results for 2020, based on excess deaths, are somewhat in line with Santos *et al.* [20], with 18.4% against 13.4% more extra deaths in males and females, respectively. However, for 2021, around 30% more deaths were estimated for both sexes. Our analysis pointed out more prominent female deficit deaths only during the plateau of the first wave, an indication that in 2020, women might have been able to engage better in social distancing and avoided contamination by other infections. That could be a consequence of the well-known vulnerable forms of employment (informal self-employing, housework and children care responsibilities) females share, mainly in South American countries [58].

The pandemic impacted age groups differently as expected and shown in other studies [20, 57, 59, 61] with a larger impact for people aged between 40 to 79 years old and 20 or older, in 2020 and 2021, respectively. The group younger than 20 showed deficit deaths in both years. In contrast, for the elderly, COVID-19 surpassed excess deaths, mainly in the first year, meaning fewer deaths from other causes. Some high-income countries also showed deficit death for youngsters [62]. Our deficit death estimate (−6.5%) is very close to that from the previous study (−7.2%) [20], and we believe the explanations there are very plausible. Although the country did not apply strict lockdown, schools were closed, and outdoor and trip activities were reduced, contributing to lower exposure to external injuries and infections.

Our results showed an exceeded excess of deaths from COVID-19 in all race/color groups, except White, who had a deficit of deaths during part of the first wave. For indigenous and East Asian descendants, the results are unclear, in part because of their sparsity (these groups represent 1.5% of the Brazilian population). We did not find other studies with Brazilian data that investigated indigenous and East Asian descendants separately, and our analysis portrayed unexpected patterns, especially for East Asians. For the indigenous people, the media repercussions on the high mortality during the first wave possibly contributed to actions for providing better care, including the priority of vaccination in 2021. These are possible explanations for the slightly better statistics in the second year. As for East Asian descendants, we do not anticipate the results because this population generally has a better social status and most live in developed areas in Brazil. A systematic review [63] indicated that minorities, including those of Asian descent, have an increased risk of infection and of developing serious outcomes compared to the white population [63]. However, most of the evidence came from the US and UK. In other studies [64, 65], the East Asian group did not show higher mortality rates compared to the white group. However, these studies considered confirmed COVID-19 deaths of hospitalized patients and periods of low coverage (up to May 2020 and up to September 2020,

respectively). Our analyzes show that East Asian descent was affected by mortality from causes other than COVID-19, and further studies are needed to explain these findings.

As for the main race/color groups, in contrast to Santos *et al.* [20], our P-Score estimates are considerably higher for all, except for whites, but are lower than those presented in another study [14]. However, all results point in the same direction, e.g. non-whites had a higher excess of deaths in the first year. Teixeira *et al.* [14] explored the racial disparities for each region and showed that the states that contributed to the marked disparities, with better numbers for whites, were those belonging to the South and Southeast, where the share of the white population is larger and, in general, enjoy better living conditions, including access to prevention and health care.

## Strengths and limitations of the study

Our analyses, stratified by several factors, are based on the final data for 2020 and the most up-to-date 2021 data available by the time of our analysis, including information for the whole year, the broader study we are aware of. In contrast to most studies, we have examined mortality and excess deaths by factors such as death cause, sex, age, race/color and region within the country. In Brazil, race/color and geographical region may represent surrogates for socioeconomic status and availability/access to healthcare, critical factors to understanding pandemic impacts. In particular, the analyses for Indigenous and East Asian descendants revealed unexpected patterns, raising further questions for a better understanding of the COVID-19 pandemic's impacts on the Brazilian population. We used a flexible estimation method which does not depend on census data and allows explicit formulae for death forecast precision accounting for all uncertainty involved in the prediction process.

Our study has limitations. We rely on governmental data and its quality. Important issues are incomplete information mainly on race/color and death-cause misreport. Data and estimates for 2021 are preliminary and expected to change due to the delay in reporting mortality. Health state secretaries have 60 days, following the end of the month of death occurrence, to fill out the national mortality system. After checking for inconsistencies and errors, a report is sent back for corrections, with the checking repeated on two or three occasions, causing a considerable delay in the availability of definitive data. As we followed 2020's data from September 2021, when they were preliminary, to September 2022, when they were declared definite, we can say the differences were mainly related to death cause updating. The total mortality increased only 0.3% while COVID-19 death counts increased 1.5%. The most significant change was in mortality due to Ill-defined Causes. We found that about 7% of the deaths preliminarily declared as Ill-defined Causes migrated to other causes. Cardiovascular, Neoplasms and Other Diseases each had an increase of about 1%. Furthermore, we do not have predictions of specific effects for 2021, possibly underestimating baseline deaths for that year. Yet, we expect the time effect included in the modeling neutralizes the possible bias.

Concerning the missing information in the data, in special for race/color, our preliminary analysis revealed that the source of the missingness is related to a few states where Brown and Black groups' share is larger than in the country's overall population. That supports the argument that death counts for Brown and Black categories are under-represented, which can lead to under-estimated expected deaths. If the down-biases are proportional, the relative measures (P-Scores, Ratio$_{EC}$) are not too seriously biased. There are alternatives to remedy the problem of missingness, such as using some missing imputation and bias-adjustment mechanisms. In the eminence of the availability of the 2022 census, some promising mechanisms might be devised.

Lastly, generalizations of our findings to other countries, even from South America, are not possible due to the heterogeneity and intrinsic characteristics of the Brazilian population.

## Conclusion

Our work provides a detailed and robust assessment of the direct and indirect impact of the COVID-19 pandemic, adding considerably to the understanding of the dimension of the tragedy the Brazilian population experienced. Given the legacy of the country's health system, the COVID-19 pandemic is a sad example of a lost opportunity for national coordination involving the federal government, states and municipalities to fight and mitigate the pandemic effects.

Our stratified analyses highlight the heterogeneous impacts across the country and socio-demographic levels and, once more, show the perverse inequalities the Brazilian population endures. More detailed investigations are possible, for example, stratifying by socio-demographic factors and causes of death within each state, leading to valuable information for more locally focused measures for pandemic mitigation.

High estimates of excess deaths classified as other causes, e.g. Ill-defined Causes, flag the unpreparedness of the system for correct diagnosis and highlight the need for investments in research, products and personnel to respond to the emergence of new diseases.

Readily tools for assessing excess deaths are helpful for the detection of mortality due to any atypical phenomena, even under circumstances of cause misreporting. One limitation is the delays involved in the death recording process. A system that quickly amends the results as data are updated would be of great relevance to complement epidemiological surveillance.

## Supporting information

**S1 File. Technical methodological details.**
(PDF)

**S2 File. Application of the step-by-step approach for the baseline model.**
(PDF)

**S1 Table. Parameter estimates, standard errors (SE) and random effect predictions for the year 2020 (RE), for the LMMs referring to the fittings in Figs 2 and 3 and Table 2 of the paper.**
(PDF)

**S2 Table. Reported, expected and estimated excess/deficit total deaths by state, accumulated for two periods, weeks 10–53, 2020 and weeks 1–52, 2021.**
(PDF)

**S3 Table. Parameter estimates, standard errors (SE) and random effect predictions for the year 2020 (RE), for the LMMs referring to the fittings in Figs 5 and 6 and Table 3 of the paper.**
(PDF)

**S1 Fig. Diagnostic graph analysis for assumptions of the fitted LMM for all-cause deaths in Brazil (Eq 4).** Standardized marginal residuals against marginal fitted response, standardized conditional residuals against predicted response, Normal probability plot for standardized least confounded conditional residual, Chi-square probability plot for Mahalanobis distance and Modified Lesaffre-Verbeck measure index plot.
(EPS)

**S2 Fig. Weekly all-cause mortality in the North Region states, Brazil, from week 1, 2015 to week 52, 2021.** Recorded and baseline mortality forecast by the LMM and the forecast plus observed COVID-19 deaths including 95% prediction intervals for 2020 and 2021 (shaded areas).
(EPS)

**S3 Fig. Weekly all-cause mortality in the Northeast Region states, Brazil, from week 1, 2015 to week 52, 2021.** Recorded and baseline mortality forecast by the LMM and the forecast plus observed COVID-19 deaths including 95% prediction intervals for 2020 and 2021 (shaded areas).
(EPS)

**S4 Fig. Weekly all-cause mortality in the Southeast Region states, Brazil, from week 1, 2015 to week 52, 2021.** Recorded and baseline mortality forecast by the linear mixed model and the forecast plus observed COVID-19 deaths including 95% prediction intervals for 2020 and 2021.
(EPS)

**S5 Fig. Weekly all-cause mortality in the South Region states, Brazil, from week 1, 2015 to week 52, 2021.** Recorded and baseline mortality forecast by the linear mixed model and the forecast plus observed COVID-19 deaths including 95% prediction intervals for 2020 and 2021.
(EPS)

**S6 Fig. Weekly all-cause mortality in the Central-West Region states, Brazil, from week 1, 2015 to week 52, 2021.** Recorded and baseline mortality forecast by the linear mixed model and the forecast plus observed COVID-19 deaths including 95% prediction intervals for 2020 and 2021.
(EPS)

**S7 Fig. Weekly mortality for the race/color category White in Brazil, stratified by death cause, from week 1, 2015 to week 52, 2021.**
(EPS)

**S8 Fig. Weekly mortality for the race/color category Brown in Brazil, stratified by death cause, from week 1, 2015 to week 52, 2021.**
(EPS)

**S9 Fig. Weekly mortality for the race/color category Black in Brazil, stratified by death cause, from week 1, 2015 to week 52, 2021.**
(EPS)

**S10 Fig. Weekly mortality for the race/color category East Asian descendants in Brazil, stratified by death cause, from week 1, 2015 to week 52, 2021.**
(EPS)

**S11 Fig. Weekly mortality for the race/color category Indigenous in Brazil, stratified by death cause, from week 1, 2015 to week 52, 2021.**
(EPS)

## Author Contributions

**Conceptualization:** Saditt Rocio Robles Colonia, Luzia Aparecida Trinca.

**Data curation:** Luzia Aparecida Trinca.

**Formal analysis:** Saditt Rocio Robles Colonia, Lara Morena Cardeal, Luzia Aparecida Trinca.

**Investigation:** Saditt Rocio Robles Colonia, Lara Morena Cardeal, Rogério Antonio de Oliveira, Luzia Aparecida Trinca.

**Methodology:** Saditt Rocio Robles Colonia, Lara Morena Cardeal, Luzia Aparecida Trinca.

**Software:** Saditt Rocio Robles Colonia, Lara Morena Cardeal, Luzia Aparecida Trinca.

**Supervision:** Rogério Antonio de Oliveira, Luzia Aparecida Trinca.

**Visualization:** Saditt Rocio Robles Colonia, Lara Morena Cardeal, Luzia Aparecida Trinca.

**Writing – original draft:** Saditt Rocio Robles Colonia, Lara Morena Cardeal, Rogério Antonio de Oliveira, Luzia Aparecida Trinca.

**Writing – review & editing:** Saditt Rocio Robles Colonia, Lara Morena Cardeal, Rogério Antonio de Oliveira, Luzia Aparecida Trinca.

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
