## [Decision Letter · Decision Letter 0]

14 Sep 2022

PONE-D-22-20908Assessing COVID-19 pandemic excess deaths in Brazil: years 2020 and 2021

PLOS ONE

Dear Dr. Trinca,

Thank you for submitting your manuscript to PLOS ONE. After careful consideration, we feel that it has merit but does not fully meet PLOS ONE’s publication criteria as it currently stands. Therefore, we invite you to submit a revised version of the manuscript that addresses the points raised during the review process.

Based on the reviwers detailed and careful comments and my own reading, I believe the paper would benefit by revising several points made by the reviewers. Please, see detaild comments. More specifically: 

Test alternative models, such as GEE, and discuss the differences in the resultsFocus on focus on the modeling strategy instead. Perhaps you could explain it more deeply, instead of explaining the regression modelCopyediting - some parts of the paper are a little bit confusing The mathematical notation is very confusingFuther discuss expected delay in the reporting mortality 2021 dataIntroduction should brief and specific description on Brazilian epidemic contextinformation on missing data to the important covariates such as race/color are unavailable. In which direction this point may have raised interpretative distortions? add the key aspects on the study limitations and strong pointPlease submit your revised manuscript by Oct 29 2022 11:59PM. If you will need more time than this to complete your revisions, please reply to this message or contact the journal office at plosone@plos.org. Please include the following items when submitting your revised manuscript:A rebuttal letter that responds to each point raised by the academic editor and reviewer(s). You should upload this letter as a separate file labeled 'Response to Reviewers'.A marked-up copy of your manuscript that highlights changes made to the original version. You should upload this as a separate file labeled 'Revised Manuscript with Track Changes'.An unmarked version of your revised paper without tracked changes. You should upload this as a separate file labeled 'Manuscript'.

We look forward to receiving your revised manuscript.

Kind regards,

Bernardo Lanza Queiroz, Ph.D

Academic Editor

PLOS ONE

Journal Requirements:

"The first and second authors acknowledge research scholarships from Coordena¸c˜ao de Aperfei¸coamento de Pessoal de N´ıvel Superior – Brasil (CAPES) – Finance Code 001." 

 "The authors received no specific funding for this work."

3. We note that Figures 3 and 4 in your submission contain [map/satellite] images which may be copyrighted. All PLOS content is published under the Creative Commons Attribution License (CC BY 4.0), which means that the manuscript, images, and Supporting Information files will be freely available online, and any third party is permitted to access, download, copy, distribute, and use these materials in any way, even commercially, with proper attribution. For these reasons, we cannot publish previously copyrighted maps or satellite images created using proprietary data, such as Google software (Google Maps, Street View, and Earth). For more information, see our copyright guidelines: http://journals.plos.org/plosone/s/licenses-and-copyright.

a) You may seek permission from the original copyright holder of Figures 3 and 4 to publish the content specifically under the CC BY 4.0 license.  

Reviewers' comments:

Reviewer's Responses to Questions

**Comments to the Author**

1. Is the manuscript technically sound, and do the data support the conclusions?

Reviewer #1: Yes

Reviewer #2: Yes

2. Has the statistical analysis been performed appropriately and rigorously? 

Reviewer #1: I Don't Know

Reviewer #2: Yes

3. Have the authors made all data underlying the findings in their manuscript fully available?

Reviewer #1: Yes

Reviewer #2: No

4. Is the manuscript presented in an intelligible fashion and written in standard English?

Reviewer #1: Yes

Reviewer #2: No

5. Review Comments to the Author

Reviewer #1: From a general point view, this is an interesting and well written paper. Even so, I am not all convinced that the estimates are sufficiently reliably due to the expected delay in the reporting mortality 2021 data in Brazil, even so the authors may wish to consider the following comments.

Overall, the introduction section would benefit from a brief and specific description on Brazilian epidemic context, mainly scientific denialism and its consequences in public health terms, such as in regard to the avoidable deaths and its relationship with excess deaths. In the following introduction sentence “By the end of 2021, Brazil’s coronavirus disease (COVID-19) death toll was 619 334 (11.4% of the world), putting the country among the most affected, behind the USA only”, instead overall mortality in December/2021 would be useful changing for the overall mortality in the end of August/2022.

Details on the excess deaths approach would be better suited in the methods (or discussion) section because there are already a large amount studies on the topic both national and international Covid-19 pandemic literature. Instead, the authors may address to the readers highlighting the pandemic impact on the mortality profile around the world and/or Brazil and adding one or more hypothesis into introduction section in regard expected results to the general or specific mortality causes of death.

In “Materials and methods” section, specifically in the following sentence “We obtained the publicly available data (2015-2021) from the Mortality Information 58 System (SIM) [22], Ministry of Health, Brazil Government, on 7 March 2022” it is not sufficiently clear if the 2020 mortality data are officially considered finalized. Whereas the 2021 data were made available preliminarily is crucial updating the dataset with the last version (June 2022).

In the results section the information on missing data to the important covariates such as race/color are unavailable. In which direction this point may have raised interpretative distortions?

There is a clear unbalance between results and discussion section, showing that substantial data portion not was discussed by the authors.

Finally, is lacking to add the key aspects on the study limitations and strong points. For example, delay deaths notification, underreporting in poorest Brazilian regions, changes in the coding of underlying causes and the dates of deaths, impact of the different modeling approaches to estimate excess deaths, hard interpretation of the disaggregated analyses to the youngest people or also evaluating data on excess deaths in different pandemic stages.

Reviewer #2: Major changes

1. About the model, the author says that one criticism of LMM is the assumption of normality which is solved using weekly data. Why do not use Generalized Estimating Equations instead? The GEE models do not assume normality of data. Which could be the main differences in the results?

2. I strongly recommend eliminating all the explanation of LMM. Such models are very well known, and the aim of the paper is not to develop a new statistical procedure but to apply it. The authors could focus on the modeling strategy instead. Perhaps you could explain it more deeply.

3. The rest of the mathematical description of the model could be relocated at the Appendix. It is unnecessary to explain it with this detail because, as I stated above, it is not a paper on a new methodological development.

4. Check in the literature if “Brown” is a category of race and justify it.

5. Race is not the same as skin color according with recent literature. Please review the proper literature about it, for instance:

Jablonski, NG. Skin color and race. Am J Phys Anthropol. 2021; 175: 437– 447. https://doi.org/10.1002/ajpa.24200

Minor changes

1. Copyediting

2. I recommend to review and include following references:

Lima, E.E.C., Vilela, E.A., Peralta, A. et al. Investigating regional excess mortality during 2020 COVID-19 pandemic in selected Latin American countries. Genus 77, 30 (2021). https://doi.org/10.1186/s41118-021-00139-1

L B Nucci, C C Enes, F R Ferraz, I V da Silva, A E M Rinaldi, W L Conde, Excess mortality associated with COVID-19 in Brazil: 2020–2021, Journal of Public Health, 2021

3. I strongly recommend to review this paper:

Palacio-Mejía, L. et al. (2022). Leading causes of excess mortality in Mexico during the COVID-19 pandemic 2020–2021: A death certificates study in a middle-income country, The Lancet Regional Health – Americas, https://doi.org/10.1016/j.lana.2022.100303

4. How the results and methods of this paper are different from https://ourworldindata.org/excess-mortality-covid ?

5. The quotation is quite confusing.

6. Define the acronyms of the states at the Appendix or perhaps the would be included with the maps.

7. The mathematical notation is very confusing, and in some parts the variables, functions and rank notation are not defined. I strongly recommend to check it and simplify it.

6. PLOS authors have the option to publish the peer review history of their article (what does this mean?). If published, this will include your full peer review and any attached files.

Reviewer #1: **Yes: **Jesem Orellana

Reviewer #2: No

---

## [Author Response · Author response to Decision Letter 0]

15 Dec 2022

We thank the reviewers for your time and effort in reviewing our manuscript. The feedback has been invaluable in improving the content and presentation of the paper. Our point-by-point responses are given below:

Responses to the Editor

1. Test alternative models, such as GEE, and discuss the differences in the results

We have tested a GEE approach and included some minimal comparisons. The GEE results show a tendency to underestimate expected deaths, but our great concern is accounting for the uncertainties in the forecasts. We do not raise this element in the paper because it is already a long paper, and we felt the issue requires much deeper treatment, out of the aims of our paper. We extend this point a bit more in the responses for the reviewers.

2. Focus on the modeling strategy instead. Perhaps you could explain it more deeply, instead of explaining the regression model

Done. We have removed technical details from the paper and transferred the LMM theory, estimation of excess deaths, etc... to the Support Information.

3. Copyediting - some parts of the paper are a little bit confusing

We tried to be more direct.

4. The mathematical notation is very confusing

Simplifications were applied.

5. Further discuss expected delay in the reporting mortality 2021 data

We discussed the issue and have also included details based on preliminary and final 2020 data, so that some guesses of possible impacts of delay for 2021 can be made.

6. Introduction should brief and specific description on Brazilian epidemic context

Done.

7. Information on missing data to the important covariates such as race/color are unavailable. In which direction this point may have raised interpretative distortions?

Done. We investigated the missing patterns and tried tracking their source. Based on that, we included, in the discussion, our view of the biases direction.

8. Add the key aspects on the study limitations and strong point

Done.

\\vspace{1cm}

\\textbf{Responses to the Reviewers

1. Is the manuscript technically sound, and do the data support the conclusions?

%The manuscript must describe a technically sound piece of scientific research with data that supports the conclusions. Experiments must have been conducted rigorously, with appropriate controls, replication, and sample sizes. The conclusions must be drawn appropriately based on the data presented.

Reviewer \\#1: Yes

Reviewer \\#2: Yes

2. Has the statistical analysis been performed appropriately and rigorously?

Reviewer \\#1: I Don't Know

There are many alternatives for obtaining baseline forecasts during pandemic periods. The main differences between them are related to the amount of information required and the model assumptions. Our approach requires mortality data only from as large as the wanted historical death period. The model involves classical assumptions while considering correlation and heterogeneity. The biggest impact, we believe, is the explicit accounting for all uncertainties in forecasting deaths for the non-observed period. It is our opinion that quite a few methods being used do not account for all uncertainty, resulting in unreliable prediction precision.

Reviewer \\#2: Yes

3. Have the authors made all of the data underlying the findings in their manuscript fully available?

Reviewer \\#1: Yes

Reviewer \\#2: No

The data is publicly available online from the pages of the Health Ministry, Brazilian Government and we have fully referenced the site.

4. Is the manuscript presented in an intelligible fashion and written in standard English?

%PLOS ONE does not copyedit accepted manuscripts, so the language in submitted articles must be clear, correct, and unambiguous. Any typographical or grammatical errors should be corrected at revision, so please note any specific errors here.

Reviewer \\#1: Yes

Reviewer \\#2: No

We rewrote large parts and revised the whole text. We hope we improved this point.

5. Review Comments to the Author

%Please use the space provided to explain your answers to the questions above. You may also include additional comments for the author, including concerns about dual publication, research ethics, or publication ethics. (Please upload your review as an attachment if it exceeds 20,000 characters)

Reviewer \\#1: 

From a general point view, this is an interesting and well written paper. Even so, I am not all convinced that the estimates are sufficiently reliably due to the expected delay in the reporting mortality 2021 data in Brazil, even so the authors may wish to consider the following comments.

During the revision process, SIM data were updated. We performed all the analyses again and, in this revised version, present results for the updated data. Still, 2021 is not final. We discuss, in the manuscript, the expected impacts of the reporting delay of mortality for 2021.} 

Overall, the introduction section would benefit from a brief and specific description on Brazilian epidemic context, mainly scientific denialism and its consequences in public health terms, such as in regard to the avoidable deaths and its relationship with excess deaths.

We rewrote the introduction and implemented the suggestion.

In the following introduction sentence “By the end of 2021, Brazil’s coronavirus disease (COVID-19) death toll was 619 334 (11.4\\% of the world), putting the country among the most affected, behind the USA only”, instead overall mortality in December/2021 would be useful changing for the overall mortality in the end of August/2022.

Done.

Details on the excess deaths approach would be better suited in the methods (or discussion) section because there are already a large amount studies on the topic both national and international Covid-19 pandemic literature. Instead, the authors may address to the readers highlighting the pandemic impact on the mortality profile around the world and/or Brazil and adding one or more hypothesis into introduction section in regard expected results to the general or specific mortality causes of death.

We addressed these suggestions.

In “Materials and methods” section, specifically in the following sentence “We obtained the publicly available data (2015-2021) from the Mortality Information 58 System (SIM) [22], Ministry of Health, Brazil Government, on 7 March 2022” it is not sufficiently clear if the 2020 mortality data are officially considered finalized. Whereas the 2021 data were made available preliminarily is crucial updating the data set with the last version (June 2022).

Done.

In the results section the information on missing data to the important covariates such as race/color are unavailable. In which direction this point may have raised interpretative distortions?

Done.

There is a clear unbalance between results and discussion section, showing that substantial data portion not was discussed by the authors.

Finally, is lacking to add the key aspects on the study limitations and strong points. For example, delay deaths notification, underreporting in poorest Brazilian regions, changes in the coding of underlying causes and the dates of deaths, impact of the different modeling approaches to estimate excess deaths, hard interpretation of the disaggregated analyses to the youngest people or also evaluating data on excess deaths in different pandemic stages.

We addressed these points, although we believe just touching the impact of different estimation methods. That would be quite a deep study, we think.

Reviewer \\#2: Major changes

1. About the model, the author says that one criticism of LMM is the assumption of normality which is solved using weekly data. Why do not use Generalized Estimating Equations instead? The GEE models do not assume normality of data. Which could be the main differences in the results?

We have included a brief comparison of LMM and a GEE approach.

2. I strongly recommend eliminating all the explanation of LMM. Such models are very well known, and the aim of the paper is not to develop a new statistical procedure but to apply it. The authors could focus on the modeling strategy instead. Perhaps you could explain it more deeply.

Done.

3. The rest of the mathematical description of the model could be relocated at the Appendix. It is unnecessary to explain it with this detail because, as I stated above, it is not a paper on a new methodological development.

We tried to present minimal mathematical details in the paper and allocated the details in the Supporting Information.

4. Check in the literature if “Brown” is a category of race and justify it.

Brown ("Parda" in Portuguese) is a category of multidimensional criteria supposed to collect ethnic-racial information, included in official statistics in Brazil. The question is "Sua cor ou raça é:" (Your color or race is:). The available categories are "Branca" (White), "Preta" (Black), "Parda" (Brown), "Amarela" (Asian) and "Indigena" (Indigenous). The interpretation of any answer should be "as one perceives oneself" in the context of the Brazilian population. In the case of mortality data, a family member answers the question, if possible. So, even if we accept that race exists, neither "White" is a category of race in our data.} 

5. Race is not the same as skin color according with recent literature. Please review the proper literature about it, for instance:

Jablonski, NG. Skin color and race. Am J Phys Anthropol. 2021; 175: 437– 447. https://doi.org/10.1002/ajpa.24200

Thank you for bringing to attention this rather complex theme. This point is the one we felt the most difficult to address and we must admit the discussion is out of our expertise. What we can explain is that we used race/color (we opt now to remove "skin" because it is not as it was asked as detailed above) to account for a multi-dimensional factor that diversifies the Brazilian population, in a social context instead of a biological marker. It is the only information available in the mortality data that allows exploration of the pandemic impact in groups with different hardships, which perhaps is linked to the inhuman past that we cannot erase or pretend did not happen. In Brazil, the information on this factor is considered crucial to generate subsidies and indicators that favor decision-making policies (governmental and non-governmental) for resource allocation, and implementation of programs to reduce inequalities in opportunities in general, including healthcare. Are there better approaches? We truly have no idea for this country. Meanwhile, we hope that the stratified analysis we present can show, at least, useful information for improving transparency of the pandemic's impacts on our most diverse population. We tried to add some context for this factor in the paper.

Minor changes

1. Copyediting

Done.

2. I recommend to review and include following references:

Lima, E.E.C., Vilela, E.A., Peralta, A. et al. Investigating regional excess mortality during 2020 COVID-19 pandemic in selected Latin American countries. Genus 77, 30 (2021). https://doi.org/10.1186/s41118-021-00139-1

L B Nucci, C C Enes, F R Ferraz, I V da Silva, A E M Rinaldi, W L Conde, Excess mortality associated with COVID-19 in Brazil: 2020–2021, Journal of Public Health, 2021

Done. Thanks for pointing these to us.

3. I strongly recommend to review this paper:

Palacio-Mejía, L. et al. (2022). Leading causes of excess mortality in Mexico during the COVID-19 pandemic 2020–2021: A death certificates study in a middle-income country, The Lancet Regional Health – Americas, 

https://doi.org/10.1016/j.lana.2022.100303

Done.

4. How the results and methods of this paper are different from https://ourworldindata.org/excess-mortality-covid ?

As we understand, ourworldindata.org presents results from a few methods, for example, Karlinsky and Koback (2021) and the so-called WHO method. The first method estimates the baseline using a regression model with smoothing-spline-based weekly effects (normality assumed). The second one uses a GLM (Generalized Linear Model), with Negative Binomial or Poisson distributions. Both methods do not consider the correlation among death counts over time. We believe the different methods impact mainly the precision of forecasts. For the first method, ourworldindata.org does not present any uncertainty measure, and point forecasts are comparable to ours. The WHO method calculates precision for the forecasts based on simulations after fitting the model. It needs those simulations to really incorporate the uncertainty with respect to future} deaths because GLMs are not devised for predictions. We briefly mention this in the paper. The same issue arises for GEE. As we understand, GEEs are devised for estimating a mean profile, not for predicting a new observation. We can project the estimated equation to a future time and get a point prediction but the precision for that prediction is not reliable if we account only for the error in the parameter estimates. 

In contrast, LMM allows explicit accounting of uncertainties around point forecasts in a simpler way, just extending linear model theory formulae, no simulations or unusual approximations are required. That is because of the additive nature of random terms in LMMs. Details about this property were transferred to the Support Information in this revised version.

5. The quotation is quite confusing.

Sorry, we had difficulty interpreting this. For reference citations, we used the journal style and revised some specific points we had been using authors' names to address the work.

6. Define the acronyms of the states at the Appendix or perhaps the would be included with the maps.

We have substituted the maps for more informative graphs. Now the states' full names are displayed.

7. The mathematical notation is very confusing, and in some parts the variables, functions and rank notation are not defined. I strongly recommend to check it and simplify it.

Done.

---

## [Decision Letter · Decision Letter 1]

15 Feb 2023

PONE-D-22-20908R1Assessing COVID-19 pandemic excess deaths in Brazil: years 2020 and 2021PLOS ONE

Dear Dr. Trinca,

Thank you for submitting your manuscript to PLOS ONE. After careful consideration, we feel that it has merit but does not fully meet PLOS ONE’s publication criteria as it currently stands. Therefore, we invite you to submit a revised version of the manuscript that addresses the points raised during the review process.

We look forward to receiving your revised manuscript.

Kind regards,

Muhammad Aamir, Ph.D.

Academic Editor

PLOS ONE

Journal Requirements:

Additional Editor Comments:

The authors may incorporate the comments suggested by Reviewers 3 and 4. Secondly, it is suggested to add conclusion section along with limitations and future recommendations.

Reviewers' comments:

Reviewer's Responses to Questions

**Comments to the Author**

1. If the authors have adequately addressed your comments raised in a previous round of review and you feel that this manuscript is now acceptable for publication, you may indicate that here to bypass the “Comments to the Author” section, enter your conflict of interest statement in the “Confidential to Editor” section, and submit your "Accept" recommendation.

Reviewer #1: All comments have been addressed

Reviewer #3: (No Response)

Reviewer #4: (No Response)

2. Is the manuscript technically sound, and do the data support the conclusions?

Reviewer #1: Yes

Reviewer #3: Partly

Reviewer #4: Yes

3. Has the statistical analysis been performed appropriately and rigorously? 

Reviewer #1: Yes

Reviewer #3: No

Reviewer #4: Yes

4. Have the authors made all data underlying the findings in their manuscript fully available?

Reviewer #1: Yes

Reviewer #3: No

Reviewer #4: Yes

5. Is the manuscript presented in an intelligible fashion and written in standard English?

Reviewer #1: Yes

Reviewer #3: Yes

Reviewer #4: Yes

6. Review Comments to the Author

Reviewer #1: After careful review, I feel the revised version clarifies important points, mainly regarding the Materials and methods section, turning up the manuscript results even more robust. I am therefore in favour of the accept this new manuscript version.

Reviewer #3: The authors incorporated most of the concerns raised by the reviewers. However, still some of the work is not incorporated. i.e. (1) There is still irrelevant statements in the manuscript, for instance, "An advantage of the linear mixed model is the flexibility to capture year-trend while dealing with the correlations among death counts over time".

(2) There is other methods available in the literature as mentioned by the 2nd reviewer. Why the authors focused on LMM only besides its flexibility and relationship among the variables?

(3) The authors only displayed the tables for expected and estimated deaths, why not they showing the coefficients of LMM?

(4) Why the authors included the GLM and AR in the discussion section? is there really need to discuss it in the discussion section?

(5) The authors showed that South is less impacted while the North and Central-West had the highest in Brazil. It is suggested that also bring the possible and general reasons/factors.

(6) It is suggested that to add conclusion section along with limitation and future recommendation.

(7) Overall the paper need technical improvement, i.e. to remove irrelevant materials.

Reviewer #4: The authors analyzed COVID-19 pandemic-induced mortality in Brazil over a period of two years. Although, the paper lacks novelty but strongly addresses the epidemiological patterns in this study. The comments of previous reviewers are addressed appropriately. I recommend adding a few lines about the implications of the study for any future pandemics like COVID-19.

There are a few editorial mistakes e.g.,

1. in line 253 add “and we found..”

2. Line 85 starts with reference number 18. Please change it to author according to the refereeing guidelines.

3. I suggest changing Line 106, “Another possible approach is the application of the Generalized Estimation 105 Equations (GEE) formulation [32], which, while considering correlations among the 106 observations, does not require any distributional assumption” to “Another possible approach is the application of the Generalized Estimation 105 Equations (GEE) formulation [32], which does not require any distributional assumption when correlations among the 106 observations considered”

7. PLOS authors have the option to publish the peer review history of their article (what does this mean?). If published, this will include your full peer review and any attached files.

Reviewer #1: **Yes: **Jesem Douglas Yamall Orellana

Reviewer #3: No

Reviewer #4: **Yes: **Aisha Naeem

---

## [Author Response · Author response to Decision Letter 1]

8 Mar 2023

Responses to the Editor

In addition to our responses to specific points below, we would like to inform we have checked the reference list and:

1. We have corrected some issues, mainly concerning doi specifications.

2. We have added one reference, currently number 43 because it supports our answer to some points asked by Reviewer # 3.

3. Furthermore, we removed reference number 45 from the previous version and, instead, cited the data link on the web in the text (line 278).

4. We are not aware of any retracted work in our reference list.

Additional Editor Comments:

The authors may incorporate the comments suggested by Reviewers 3 and 4. Secondly, it is suggested to add a conclusion section along with limitations and future recommendations.

We have included a conclusion section.

Responses to Reviewers

Reviewer’s Responses to Questions

2. Is the manuscript technically sound, and do the data support the conclusions?

Reviewer #1: Yes

Reviewer #3: Partly

Our comment: We have included a web link for the aggregated data and some program code for the analyses and a conclusion section in the main text and tables with model parameters' estimates in the supplementary information.

Reviewer #4: Yes

3. Has the statistical analysis been performed appropriately and rigorously?

Reviewer #1: Yes

Reviewer #3: No

Our comment: We have performed diagnostics for the models fitted and did not find any serious violations of the models’ assumptions indicating the models are reasonable approximations for the data. We opted to show just some of the diagnostic graphs in the Supplementary Information material because of the large number of figures.

Reviewer #4: Yes

4. Have the authors made all data underlying the findings in their manuscript fully available?

Reviewer #1: Yes

Reviewer #3: No

Our comment: Data is publicly available, online, from the pages of the Health Ministry, Brazilian Government, as we had fully referenced in the References section. In this revision, we have included the links in the text, subsection Data (lines 53/54) and also, our GitHub repository with processed (aggregated) data, program code, etc (line 82).

Reviewer #4: Yes

5. Is the manuscript presented in an intelligible fashion and written in standard English?

Reviewer #1: Yes

Reviewer #3: Yes

Reviewer #4: Yes

6. Review Comments to the Author

Reviewer #1: After careful review, I feel the revised version clarifies important points, mainly regarding the Materials and methods section, turning up the manuscript results even more robust. I am therefore in favour of the accept this new manuscript version.

Thank you for your careful review and comments in both versions which helped us to improve our work.

Reviewer #3: The authors incorporated most of the concerns raised by the reviewers. However, still, some of the work is not incorporated. i.e.

(1) There are still irrelevant statements in the manuscript, for instance, ”An advantage of the linear mixed model is the flexibility to capture year-trend while dealing with the correlations among death counts over time”.

Thank you for your careful review. We have, hopefully, removed all the irrelevant parts in the text.

(2) There is other methods available in the literature as mentioned by 

the 2nd reviewer. Why the authors focused on LMM only besides its flexibility and relationship among the variables?

Yes, there are many different methods available and, initially, based on Verbeeck et al. (2021) we chose LMM. These authors pointed out some drawbacks of several methods and favored the LMM. However, based on the comments of Reviewer 2 in the first version, we did some comparisons between three methods, one of them being the method mentioned by Reviewer 2, the GEE. In Table 1 of the paper, we show that LMM and GEE perform similarly in terms of point predictions while the five-year average is much poorer. In Figure 1 we compare the fitted curves from LMM and

GEE where we can see LMM captures better the peaks. Allied with that, as explained in the text, GEE is not suited for future predictions (and we want to predict death counts had the pandemic not occurred). That is because the method does not consider errors associated with future points in time, resulting in practically null standard errors for the predictions. In our opinion, that can mislead conclusions on excess deaths. Another method that would be sound is GLMM but it requires simulations to properly account for the uncertainty around predictions. Based on that, we chose to use LMM which, explicitly, incorporates all uncertainty around predictions. Details on standard error calculations are presented in S1 File.

(3) The authors only displayed the tables for expected and estimated deaths, why not they showing the coefficients of LMM?

New tables with all the parameter estimates are provided in the Supplemental Information, S1 Tab and S3 Tab.

(4) Why the authors included the GLM and AR in the discussion section? is there really need to discuss it in the discussion section?

We have removed some details of these methods in the discussion but found it informative to keep their names because the five-year average tends to overestimate excess. Besides, the completest previous work on excess deaths in Brazil (reference [6]) used GLM, and we felt it would be flawed not to include it in the discussion.

(5) The authors showed that the South is less impacted while the North and Central-West had the highest in Brazil. It is suggested that also bring the possible and general reasons/factors. We added some possible explanations for the heterogeneity across the country. We have included, in Figure 4 of the paper, the Human Development Index and vaccination coverage as surrogates for life quality, education, and economic and health capacity of each state in an attempt to draw some possible explanations for the heterogeneous impact. However, it is not possible, with our data and analysis, to really account for the factors behind the heterogeneous impact across the country.

(6) It is suggested that to add conclusion section along with limitation and future recommendation.

The Conclusion section is included.

(7) Overall the paper need technical improvement, i.e. to remove irrelevant materials.

Done.

Reviewer #4: The authors analyzed COVID-19 pandemic-induced mortality in Brazil over a period of two years. Although, the paper lacks novelty but strongly addresses the epidemiological patterns in this study. The comments of previous reviewers are addressed appropriately. I recommend adding a few lines about the implications of the study for any future pandemics like COVID-19.

Thank you for your careful review. We included a conclusion section.

There are a few editorial mistakes e.g.,

1. in line 253 add “and we found..”

Fixed.

2. Line 85 starts with reference number 18. Please change it to author according to the refereeing guidelines.

Done.

3. I suggest changing Line 106, “Another possible approach is the application of the Generalized Estimation Equations (GEE) formulation [32], which, while considering correlations among the observations, does not require any distributional assumption” to “Another possible approach is the application of the Generalized Estimation Equations (GEE) formulation [32], which does not require any distributional assumption when correlations among the 106 observations considered”

Thanks for the suggestion. To express the proper meaning we have rephrased it to ”Another possible approach is the application of the Generalized Estimation Equations (GEE) formulation [32], which does not require any distributional assumption and allows correlations among the observations.”

---

## [Decision Letter · Decision Letter 2]

24 Apr 2023

PONE-D-22-20908R2Assessing COVID-19 pandemic excess deaths in Brazil: years 2020 and 2021PLOS ONE

Dear Dr. Trinca,

Thank you for submitting your manuscript to PLOS ONE. After careful consideration, we feel that it has merit but does not fully meet PLOS ONE’s publication criteria as it currently stands. Therefore, we invite you to submit a revised version of the manuscript that addresses the points raised during the review process.

We look forward to receiving your revised manuscript.

Kind regards,

Ivan Filipe de Almeida Lopes Fernandes, Ph.D.

Academic Editor

PLOS ONE

Journal Requirements:

Additional Editor Comments:

Minor changes

1) Abstract

From "… from the Health Ministry, the Brazilian Government, to fit linear mixed models for forecasting …

To "… from the Brazilian Health Ministry for forecasting…."

2) Introduction – pg 2, line 10

There is no delegation of the Supreme Court for states and municipalities. Supreme Court has decided that local governments should opt for more severe NPIs than the Federal Government.

See "Abrucio, F. L., Grin, E., & Segatto, C. I. (2021a). Brazilian Federalism in the Pandemic. In Peters, G., Abrucio, F. & Grin, E (ed). American Federal Systems and COVID-19. Emerald Publishing Limited" for a full description of the Supreme Court's role during the pandemic.

3) Introduction – pg 2, line 23

From “…the huge…”

To “… huge…”

4) Methods – pg 5, line 106

From ". Yet, as far we understand, it .."

To "Yet, it.."

5) Methods – pg 5, line 114

From ". Its drawback, in our opinion, is .."

To "Its drawback is."

The affirmation should be based on a literature reference on the methodology.

6) All links should be changed for References. The links should be presented in the Reference list.

7) Estimated the effect of Asian and Indigenous groups separated.

I understand that they are both small groups, but the effects of COVID-19 on these two groups are different. Asians have higher social status and Indigenous groups were heavily affected by government policies during the pandemic.

8) The text should be rewrite whenever numbers is used in the place of reference Names.

For example

Pg. 22, line 502-503

From “… also pointed out in [19], [26] and [44], although [26] and [44] used different deaths-cause grouping….”

To “…also pointed out in other studies [19, 26, 44], although some of then [26, 44] used differen deaths-cause grouping…”t

Reviewers' comments:

Reviewer's Responses to Questions

**Comments to the Author**

1. If the authors have adequately addressed your comments raised in a previous round of review and you feel that this manuscript is now acceptable for publication, you may indicate that here to bypass the “Comments to the Author” section, enter your conflict of interest statement in the “Confidential to Editor” section, and submit your "Accept" recommendation.

Reviewer #3: (No Response)

Reviewer #4: All comments have been addressed

2. Is the manuscript technically sound, and do the data support the conclusions?

Reviewer #3: Yes

Reviewer #4: Yes

3. Has the statistical analysis been performed appropriately and rigorously? 

Reviewer #3: Yes

Reviewer #4: Yes

4. Have the authors made all data underlying the findings in their manuscript fully available?

Reviewer #3: Yes

Reviewer #4: Yes

5. Is the manuscript presented in an intelligible fashion and written in standard English?

Reviewer #3: Yes

Reviewer #4: Yes

6. Review Comments to the Author

Reviewer #3: The authors incorporated most of the concerns raised the reviewers. However, limitations and future recommendations are still not provide. Please provide these well.

Reviewer #4: I have no further comments on the manuscript. The authors have addressed all the comments appropriately.

7. PLOS authors have the option to publish the peer review history of their article (what does this mean?). If published, this will include your full peer review and any attached files.

Reviewer #3: No

Reviewer #4: No

---

## [Author Response · Author response to Decision Letter 2]

4 May 2023

Thank you for your decision letter of April 24, 2023, and for the opportunity to revise our manuscript. My co-authors and I are pleased to submit our revised manuscript titled “Assessing COVID-19 pandemic excess deaths in Brazil: years 2020 and 2021” (Ref: PONE-D-22-20908R2) for your reconsideration for publication in PLOS ONE.

We have revised our manuscript according to all yours and the reviewers’ comments.

In addition to our responses to specific points below, we would like to

inform we have checked the reference list and:

1. We have added a few references accordingly to our revision.

2. We are not aware of any retracted work in our reference list.

Additional Editor Comments:

Minor changes

1) Abstract

From ”. . . from the Health Ministry, the Brazilian Government, to fit linear

mixed models for forecasting . . . To ”. . . from the Brazilian Health Ministry for forecasting. . . .”

Done.

2) Introduction – pg 2, line 10

There is no delegation of the Supreme Court for states and municipalities.

Supreme Court has decided that local governments should opt for more severe NPIs than the Federal Government. See ”Abrucio, F. L., Grin, E., &

Segatto, C. I. (2021a). Brazilian Federalism in the Pandemic. In Peters,

G., Abrucio, F. & Grin, E (ed). American Federal Systems and COVID-19.

Emerald Publishing Limited” for a full description of the Supreme Court’s

role during the pandemic.

Thank you for clarifying this issue to us. We have rewritten the

text and included references.

3) Introduction – pg 2, line 23 From “. . . the huge. . . ” To “. . . huge. . . ”

Done.

4) Methods – pg 5, line 106 From ”. Yet, as far we understand, it ..” To

”Yet, it..”

Done.

5) Methods – pg 5, line 114 From ”. Its drawback, in our opinion, is

..” To ”Its drawback is.” The affirmation should be based on a literature

reference on the methodology.

Done. We revised this part and added references.

6) All links should be changed for References. The links should be presented in the Reference list.

Done. Unfortunately, in the previous revision, we decided to move

the links to the text because a referee kept insisting we did not provide access to the data, although the links were in the reference

list. Now, in this version, the links are back to the Reference list.

We hope we got it right this time.

7) Estimated the effect of Asian and Indigenous groups separated. I

understand that they are both small groups, but the effects of COVID-19

on these two groups are different. Asians have higher social status and

Indigenous groups were heavily affected by government policies during the

pandemic.

Done. Thank you for the suggestion. The dis-aggregated analysis show an unexpected pattern for Asians and we added plots of

deaths by cause of death, for each group to try understanding the

patterns. Those graphs are in the Supporting Information.

8) The text should be rewrite whenever numbers is used in the place of

reference Names. For example Pg. 22, line 502-503 From “. . . also pointed

out in [19], [26] and [44], although [26] and [44] used different deaths-cause

grouping. . . .” To “. . . also pointed out in other studies [19, 26, 44], although

some of then [26, 44] used differen deaths-cause grouping. . . ”

Done.

Responses to Reviewers

Reviewer’s Responses to Questions

The Reviewers responses for Itens 1 to 5 were ”Yes”, so thank

them for recognizing the improvements in our manuscript.

6. Review Comments to the Author

Please use the space provided to explain your answers to the questions

above. You may also include additional comments for the author, including concerns about dual publication, research ethics, or publication ethics.

Reviewer #3: The authors incorporated most of the concerns raised the

reviewers. However, limitations and future recommendations are still not

provide. Please provide these well.

In this version, we have described the strengths and limitations

of our work in a subsection within the Discussion. Some recommendations are outlined in the Conclusions.

Reviewer #4: I have no further comments on the manuscript. The authors have addressed all the comments appropriately.

Thank you!

---

## [Editor Report · Decision Letter 3]

11 May 2023

Assessing COVID-19 pandemic excess deaths in Brazil: years 2020 and 2021

PONE-D-22-20908R3

Dear Dr. Trinca,

We’re pleased to inform you that your manuscript has been judged scientifically suitable for publication and will be formally accepted for publication once it meets all outstanding technical requirements.

Kind regards,

Ivan Filipe de Almeida Lopes Fernandes, Ph.D.

Academic Editor

PLOS ONE
---

## [Editor Report · Acceptance letter]

15 May 2023

PONE-D-22-20908R3 

Assessing COVID-19 pandemic excess deaths in Brazil: years 2020 and 2021 

Dear Dr. Trinca:

I'm pleased to inform you that your manuscript has been deemed suitable for publication in PLOS ONE. Congratulations! Your manuscript is now with our production department. 

Kind regards, 

on behalf of

Dr. Ivan Filipe de Almeida Lopes Fernandes 

Academic Editor

PLOS ONE